# Learning to Think in Physics: Breaking Shortcut Learning in Scientific Diffusion via Representation Alignment

Haozhe Jia [* 1 2]   Pengyu Yin [* 2]   Wenshuo Chen [* 1]   Shaofeng Liang [1]   Lei Wang [3 4]   Bowen Tian [1 2 5]
Xiucheng Wang [6]   Nanqian Jia [7]   Yutao Yue [1 8]

## Abstract

Physics-informed diffusion models typically enforce PDE constraints only on final outputs, leaving intermediate representations unconstrained and prone to shortcut learning under shifted boundary conditions. We introduce **REPA-P**, a teacher-free, architecture-agnostic framework that aligns intermediate features with physical states using first-principles residuals. REPA-P attaches lightweight $1 \times 1$ projection heads to selected layers, decodes hidden activations into physical quantities, and applies PDE residual losses during training. These heads are discarded at inference, introducing **zero overhead**. Across four PDE tasks, including Darcy flow, topology optimization, electrostatic potential, and turbulent channel flow, REPA-P accelerates convergence by up to $2\times$, reduces physics residuals by up to $66.4\%$, and improves out-of-distribution robustness by up to $49.3\%$, with consistent gains on both U-Net and Diffusion Transformer backbones. Ablations show that supervising a small set of intermediate layers captures most benefits and complements output-level physics losses. Code is available at https://github.com/Hxxxz0/REPA-P.

## 1. Introduction

Generative models have achieved substantial progress in recent years, particularly in image synthesis (Ning et al.,

2025b; Esser et al., 2024) and video synthesis (Gupta et al., 2023). Achieving high-quality generation often relies on scalable and effective architectures, such as U-Net and DiT (Ronneberger et al., 2015; Peebles & Xie, 2023). Despite these advances, extending these successes to AI for Science remains fundamentally challenging. Unlike natural images, scientific data is governed by strict, immutable physical laws (Cuomo et al., 2022).

The fundamental problem with current data-driven approaches, even when augmented with physics constraints at the output, is that they may still be prone to **shortcut learning** (Bastek et al., 2025). This is particularly true when constraints are applied solely to the final prediction, allowing the internal network to bypass physical reasoning. By exploiting the powerful expressivity of deep networks, models essentially "memorize" spurious correlations, distribution-specific cues, and discretization artifacts of the training set to minimize the denoising objective, rather than genuinely "understanding" the underlying physical mechanisms. This is analogous to a student who rote memorizes the answers to math problems without understanding the derivation steps: they may score well on familiar questions, but fail catastrophically when the question parameters change. In the context of diffusion models, this lack of internal understanding may lead to fragility against out-of-distribution (OOD) boundary conditions and physical inconsistencies that are merely masked by superficial pattern matching.

**Hypothesis.** We hypothesize that the robustness and generalizability of scientific generative models can be significantly enhanced by **aligning intermediate representations with physical states**. While standard diffusion models operate effectively as statistical denoisers, we posit that if their latent features are structurally aligned with physical quantities (e.g., velocity, pressure), the model is better positioned to capture the underlying dynamics rather than relying on spurious correlations (Li et al., 2021). We refer to this property as **Physical Decodability**: the capability of a lightweight decoder to map latent features to state variables that satisfy governing equations. We argue that encouraging such decodability acts as a powerful inductive bias, guiding the model to internalize physical laws and thereby improv-

---

[1]The Hong Kong University of Science and Technology (Guangzhou), Guangzhou, China [2]Shandong University, Jinan, China [3]Data61/CSIRO, Australia [4]Griffith University, Brisbane, Australia [5]LimX Dynamics Technology Co., Ltd., Shenzhen, China [6]Xidian University, Xi'an, China [7]Peking University, Beijing, China [8]Institute of Deep Perception Technology, Jiangsu Industrial Technology Research Institute (JITRI), Nanjing, China. Correspondence to: Yutao Yue <yutaoyue@hkust-gz.edu.cn>.

*Proceedings of the 43rd International Conference on Machine Learning*, Seoul, South Korea. PMLR 306, 2026. Copyright 2026 by the author(s).

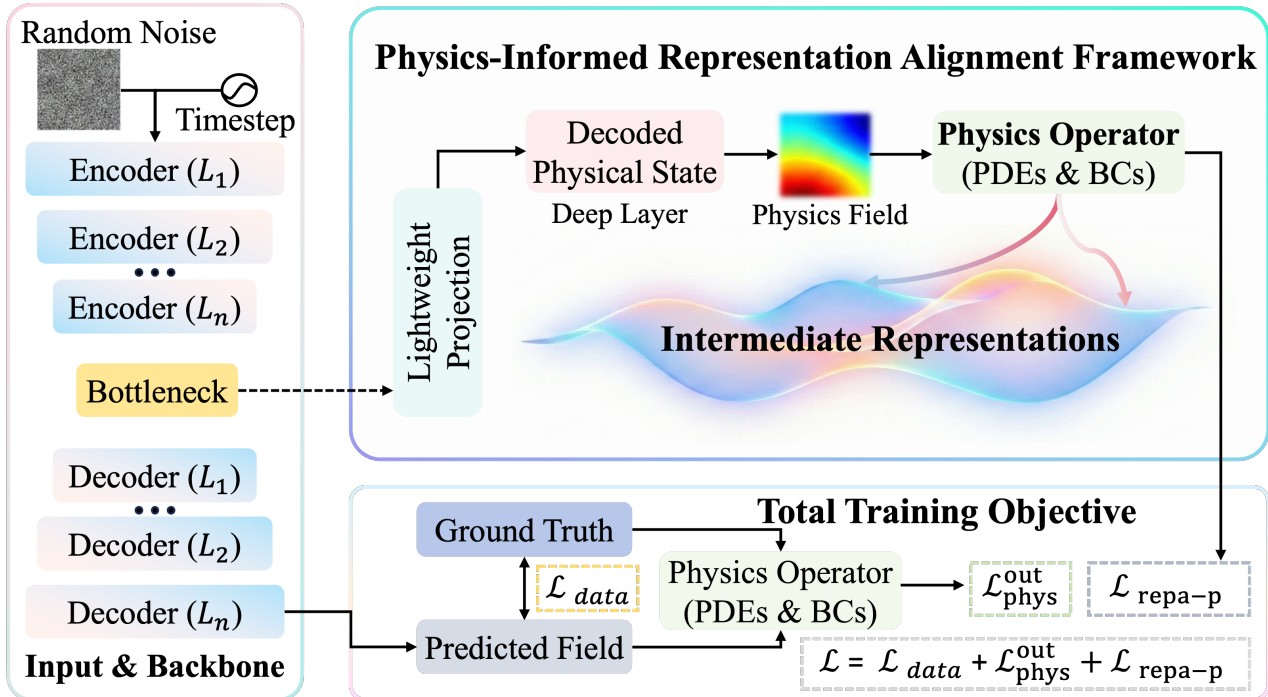

*Figure 1.* Overview of REPA-P. We decode intermediate diffusion features into physical states using lightweight projection heads and enforce PDE and boundary-condition residuals as supervision to align latent representations with valid physics.

ing both understanding and generation quality, especially in out-of-distribution scenarios (Bastek et al., 2025).

Based on this hypothesis, we propose **Physics-Informed Representation Alignment (REPA-P)**, a framework that forces the model to 'learn to think in physics' by enforcing **Physical Decodability Constraints** within the latent space. Specifically, we insert lightweight projection heads into the intermediate layers of the diffusion backbone. These heads are tasked with "translating" the high dimensional latent features into physical state variables. We then directly apply the governing Partial Differential Equations (PDEs) to these decoded states and use the PDE residuals as a supervision signal.

This mechanism fundamentally changes the learning dynamic. By calculating PDE residuals on intermediate features, we essentially force the network to "think" in the language of physics. The gradient signal effectively tells the model: *"Your internal representation must translate to valid physics."* This precludes the model from relying on statistical shortcuts or unphysical latent trajectories (Bastek et al., 2025). Crucially, this approach replaces the need for external visual teachers with the First Principles themselves, which are the most general and transferable "teachers" available in science.

Systematic experiments on standard benchmarks (e.g., Darcy Flow, Topology Optimization, Electrostatic Charge

Potential) validate our hypothesis (Bastek et al., 2025). We demonstrate that by enforcing physical decodability, the model moves beyond rote memorization. Our method not only accelerates convergence but, more importantly, achieves superior generalization on OOD tasks where standard baselines fail. This suggests that the model has successfully internalized the physical laws within its hidden layers, bridging the gap between data-driven generation and principle-based reasoning.

**Contributions.** Our main contributions are summarized as follows:

- We introduce **REPA-P**, a teacher-free, architecture-agnostic framework enforcing physical decodability on intermediate representations via lightweight $1{\times}1$ projection heads, achieving equally strong performance on both U-Net and DiT backbones.
- We demonstrate this internal alignment breaks shortcut learning via short-path gradients, compelling the model to internalize physical laws rather than memorize training patterns.
- We validate REPA-P across four PDE benchmarks covering generation and reconstruction, reducing physics residuals by up to $66.4\%$, accelerating convergence by $2\times$, with **zero inference overhead**.
- Ablations show intermediate supervision complements output-level losses, and only a small set of core layers captures most performance gains.

## 2. Related Work

**Diffusion-based generation.** Diffusion models (a.k.a. score-based generative models) have become a dominant paradigm for high-fidelity generation, tracing back to early formulations based on learning a reverse diffusion process (Sohl-Dickstein et al., 2015) and later popularized by DDPM-style denoising objectives (Ho et al., 2020) and continuous-time score/SDE views (Song et al., 2021). Subsequent work improved sampling efficiency and quality via implicit/deterministic samplers and improved training objectives (Song et al., 2020; Nichol & Dhariwal, 2021; Karras et al., 2022), as well as scalable architectures such as latent diffusion and transformer backbones (Rombach et al., 2022). For conditional generation and controllability, classifier-free guidance and control adapters are widely used (Ho & Salimans, 2022; Zhang et al., 2023). A large body of research further accelerates sampling through better solvers, distillation, and one/few-step consistency-style training (Lu et al., 2022; Salimans & Ho, 2022; Song et al., 2023; Zheng et al., 2023), and diffusion priors have also been extended to plug-and-play inverse problems (Kawar et al., 2022; Chung et al., 2024). Beyond pixel-space modeling, recent work has explored alternative representation domains such as DCT frequency space (Ning et al., 2025a) and frequency-domain consistency (Chen et al., 2025a). Orthogonal advances improve inversion stability (Chen et al., 2025b), sampling-path guidance (Li et al., 2025), and text-to-motion generation (Chen et al., 2025c; Jia et al., 2025d), with emerging applications in humanoid control (Jia et al., 2026b;a), generative evaluation (Chen et al., 2026), and latent optimization (Li et al., 2026).

**Physics-informed diffusion for scientific data.** In scientific machine learning, purely data-driven generators must respect immutable governing laws; classical physics-informed learning (e.g., PINNs) and neural operators (e.g., FNO/PINO) provide complementary routes to enforce PDE structure (Cuomo et al., 2022; Li et al., 2021; 2023). Recent physics-informed diffusion models incorporate first-principles constraints by injecting PDE residual supervision into the diffusion training/sampling process (Bastek et al., 2025; Jia et al., 2025b; Wang et al., 2025), improving physical consistency and robustness under distribution shifts. Concurrently, representation alignment for physics-informed learning has been applied to radio-map reconstruction (Jia et al., 2025c;a), where mid-layer PDE supervision improved sparse-field reconstruction. Flow matching provides an alternative generative paradigm for efficient physical field construction (Jia et al., 2025e). Unlike output-only physics losses or generic deep supervision that adds auxiliary objectives at intermediate layers, we constrain intermediate representations to be decodable into valid physical states via direct PDE gradients, which discourages shortcut learning through shortened credit assignment paths.

## 3. Method

### 3.1. Preliminaries

We employ a diffusion model (Ho et al., 2020) parameterized by $\theta$ to learn a distribution over discretized physical fields on a uniform grid $\Omega_h$. Each field $x_0 \in \mathbb{R}^{C \times H \times W}$ may contain state variables such as pressure $p$ and parameters such as permeability $K$. The model is trained to predict the clean state $x_0$ from noisy states $x_t$ (where $t \sim \mathcal{U}[1, T]$ indexes the diffusion timestep) by minimizing the denoising objective:

$$\mathcal{L}_{\text{data}}(\theta) = \mathbb{E}_{t,x_0,\epsilon} \left[ \lambda_t \| x_0 - \hat{x}_0(x_t, t) \|_2^2 \right], \quad (1)$$

where $\epsilon \sim \mathcal{N}(0, I)$ is the Gaussian noise ($I$ is the identity matrix), $\hat{x}_0(x_t, t)$ is the model prediction, and $\lambda_t$ is a weighting term derived from the noise schedule.

Physical laws are captured by a discretized residual operator $R(x_0)$ encoding PDEs and boundary conditions (BCs). For instance, in Darcy flow, $R$ approximates mass conservation $\nabla \cdot (K \nabla p) - f_s$, where $f_s$ is the source term. We adopt the virtual-observable framework (Bastek et al., 2025), modeling residuals as Gaussian variables with variance $\sigma^2(t)$ proportional to the diffusion posterior variance $\tilde{\beta}_t$:

$$R(x_0) := \begin{bmatrix} \mathcal{F}_h[x_0] \\ \mathcal{B}_h[x_0] \end{bmatrix}, \quad (2)$$

$$\sigma^2(t) = \tilde{\beta}_t = \frac{1 - \bar{\alpha}_{t-1}}{1 - \bar{\alpha}_t} \beta_t. \quad (3)$$

where $\mathcal{F}_h$ and $\mathcal{B}_h$ denote the discretized interior PDE and boundary condition operators, respectively. Here, $\beta_t$ represents the forward noise variance schedule, and $\bar{\alpha}_t = \prod_{s=1}^{t}(1 - \beta_s)$ is the cumulative noise coefficient. This variance scaling $\sigma^2(t)$ ensures that physics supervision is dynamically calibrated to the noise level, becoming stricter as the denoising process converges.

### 3.2. Physics-Informed Representation Alignment

In this section, we introduce the **Physics-Informed Representation Alignment (REPA-P)** framework. We first define the alignment mapping that projects latent features to physical space, then describe the layer-wise physics loss, and finally present the total training objective.

**Physical Decodability and Motivation.** Standard physics-informed diffusion models enforce constraints only on the final output $\hat{x}_0$ (Bastek et al., 2025). While effective for refinement, this "output-only" supervision allows the deep backbone to remain a black box, potentially leading to *shortcut learning* where the model memorizes surface statistics rather than internalizing the governing laws. **Our approach differs fundamentally** by enforcing *Physical Decodability* within the intermediate layers. We hypothesize that a robust scientific generative model should possess an internal

representation where hidden states are linearly (or lightly) decodable into valid physical quantities. By supervising these intermediate states with PDE residuals, we provide *in-situ* gradients that shorten the credit assignment path and preclude statistical shortcuts. This compels latent features to align with physical principles before the final decoding stage (see Appendix D for a formal analysis of gradient attenuation, a phenomenon also observed across domains such as text-to-motion generation (Jia et al., 2025d)).

**Alignment Mapping.** Let $\{h_\ell\}_{\ell=1}^L$ denote the hidden feature tensors at $L$ selected layers (e.g., encoder, bottleneck, decoder) of the U-Net backbone. For a mini-batch of $B$ samples, each hidden tensor has shape $h_\ell \in \mathbb{R}^{B \times C_\ell \times H_\ell \times W_\ell}$. We map each $h_\ell$ to the physical channel space via a lightweight projection head and resize the result to the output resolution $(H, W)$. We define a per-layer alignment mapping:

$$z_\ell = \Pi_\ell(h_\ell) := \mathcal{I}_\ell\big(\psi_\ell(h_\ell)\big) \in \mathbb{R}^{B \times C \times H \times W}, \quad (4)$$

where $\psi_\ell : \mathbb{R}^{C_\ell} \to \mathbb{R}^C$ is a $1 \times 1$ convolutional block (parameterized by $\phi_\ell$) that projects to the output channel dimension $C$, and $\mathcal{I}_\ell$ is a bilinear interpolation operator that resizes $(H_\ell, W_\ell) \to (H, W)$. The projected tensor $z_\ell$ acts as a proxy physical field, to which we apply the same discretized residual operator $R(\cdot)$ defined in Eq. (2).

**Total Training Objective.** For the main output, the physics loss at timestep $t$ is:

$$\mathcal{L}_{\text{phys}}^{\text{out}}(t) = \frac{1}{2} \frac{\left\| R\big(\hat{x}_0(x_t, t)\big) \right\|_2^2}{\sigma^2(t)}. \quad (5)$$

For each aligned intermediate representation $z_\ell$, we compute an analogous physics loss, averaged over a set of alignment positions $\mathcal{P} \subseteq \{1, \ldots, L\}$:

$$\mathcal{L}_{\text{repa-p}}(t) = \frac{1}{|\mathcal{P}|} \sum_{\ell \in \mathcal{P}} \underbrace{\frac{1}{2} \frac{\left\| R(z_\ell) \right\|_2^2}{\sigma^2(t)}}_{\mathcal{L}_{\text{phys}}^{(\ell)}(t)}. \quad (6)$$

The total training objective combines the data fidelity term with the output and mid-layer physics constraints:

$$\begin{aligned}
\mathcal{L}_{\text{total}}(\theta, \{\phi_\ell\}) = \mathbb{E}_{t, x_0, \epsilon}\Big[ &\mathcal{L}_{\text{data}}(\theta) + c_{\text{out}} \mathcal{L}_{\text{phys}}^{\text{out}}(t) \\
&+ c_{\text{mid}} \mathcal{L}_{\text{repa-p}}(t) \Big],
\end{aligned} \quad (7)$$

where hyperparameters $c_{\text{out}}, c_{\text{mid}} > 0$ balance the output and intermediate physics supervision.

**Numerical Stabilization: Centered Pressure.** For physical variables that are shift-invariant (e.g., pressure $p$ in incompressible flow, where the PDE $-\nabla \cdot (K \nabla p) = f$ depends only on gradients), the solution is unique only up

to an additive constant. This ambiguity can cause numerical instability during training, as the model effectively faces a null space in the optimization landscape. To resolve this, we enforce a zero-mean constraint. Let $p$ denote the pressure component of the state $x_0$, and define the spatial mean over grid $\Omega_h$ as $\bar{p} = \frac{1}{HW} \sum_{i,j} p[i, j]$, where $i, j$ index the spatial grid positions. We then evaluate the physics loss using the centered pressure:

$$p^\circ = p - \bar{p}, \qquad \mathcal{L}_{\text{phys}} \propto \left\| R(p^\circ, K) \right\|_2^2, \quad (8)$$

where the notation $\propto$ indicates that we minimize the squared residual norm (equivalently, maximizing the log-likelihood of the virtual observation). This centering ensures residuals are evaluated on a unique solution branch.

**Discussion.** The gradients produced by Eq. (6) act *in situ* on intermediate representations. Sharing the same $t$-dependent scaling $\sigma^2(t)$ across Eq. (5) and Eq. (6) aligns the training signal with the reliability of denoising steps. At inference, the projection heads $\{\psi_\ell\}$ are discarded, incurring **zero inference overhead**.

## 4. Experiments

We evaluate **REPA-P** on four PDE-governed tasks spanning distinct physical domains: *Darcy flow*, *Topology optimization*, *Electrostatic charge potential*, and *Turbulent channel flow*. These benchmarks cover unconditional generation, conditional generation, and sparse reconstruction, enabling comprehensive assessment of physics-consistent representation learning.

**Research questions.** We investigate:

**(Q1)** Does **REPA-P** accelerate the learning of physics-consistent representations?

**(Q2)** Does **REPA-P** improve overall data fitting and convergence?

**(Q3)** Does **REPA-P** enhance the generative quality of physics-informed diffusion models across different network architectures?

### 4.1. Experimental Settings

We evaluate REPA-P on four PDE-governed tasks: **Darcy flow** (steady state, $64 \times 64$), **Topology optimization** (structural compliance, $64 \times 64$), **Electrostatic charge potential** (Poisson equation, $64 \times 64$), and **Turbulent channel flow** (DNS $128 \times 48$ slice). We primarily use a U-Net backbone for comprehensive baseline comparisons, and additionally extend our evaluation to a Diffusion Transformer (DiT) architecture to demonstrate architectural generalizability. For a fair architectural comparison, the DiT backbone used 8 transformer blocks with hidden dimension 256, 8 attention

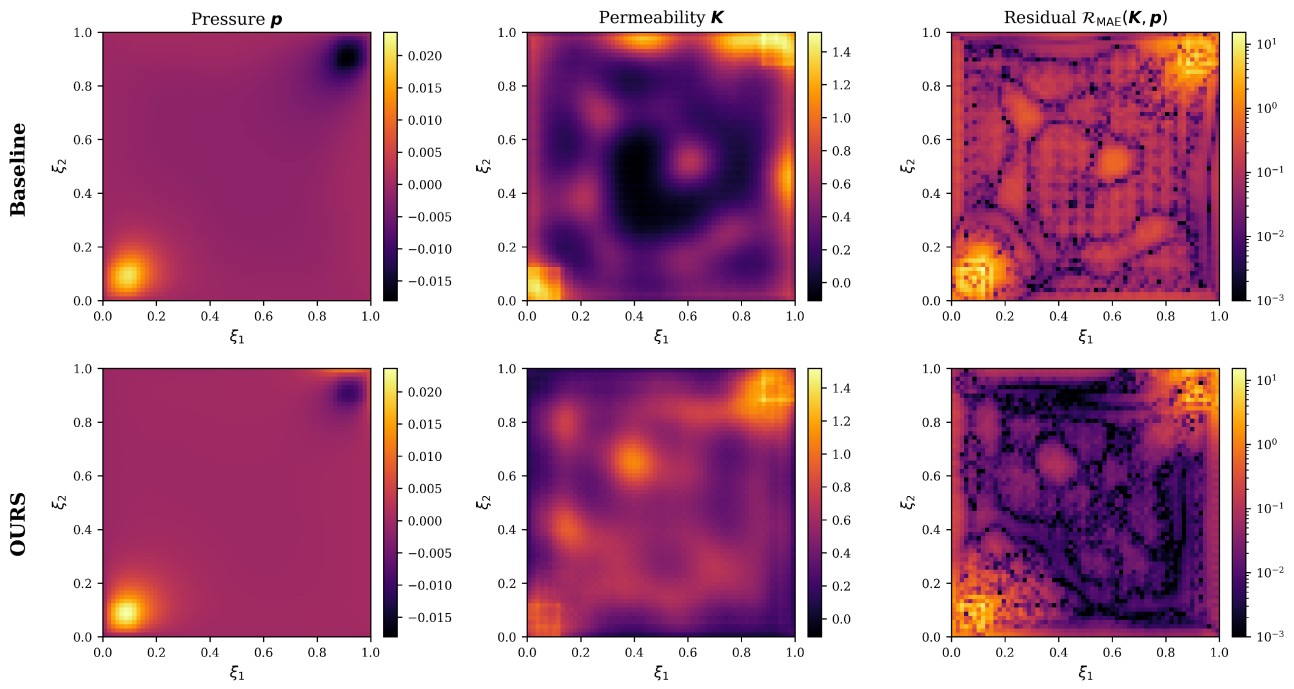

*Figure 2.* **Darcy flow qualitative comparison (Baseline vs. REPA-P).** Top: baseline diffusion; bottom: REPA-P. Each row shows (left→right) the predicted pressure $p$, the permeability field $K$, and the PDE residual $R_{\mathrm{MAE}}(K, p)$ (log scale). Compared to the baseline, REPA-P produces pressure fields that better respect the structure induced by $K$ and achieves consistently lower residuals, indicating improved satisfaction of the governing equation.

heads, patch size 4, and MLP ratio 4, yielding approximately 10M trainable parameters, comparable to the 32-channel U-Net baseline used for Darcy/turbulent tasks. We attach lightweight projection heads to intermediate layers to decode physical states and enforce PDE residuals. Detailed problem formulations, data generation procedures, architecture specifications, and training hyperparameters are provided in Appendix A. For metrics, we report physics residual MAE ($R_{\mathrm{MAE}}$), data reconstruction error (MSE/PSNR), task-specific metrics (Compliance Error for topology), and smoothing constraints for turbulence.

**Implementation.** The score networks utilize either the U-Net (Ronneberger et al., 2015) or the aforementioned DiT architecture. For Darcy flow, the network operates on $64 \times 64$ inputs and outputs that match the grid resolution, allowing the same residual evaluation used during data creation. For topology optimization, the backbone is extended with additional channels to represent structural density and loads. For the charge potential problem, the network takes the charge density $\rho$ as conditioning input and generates the electric potential $U$. For the turbulent channel flow task, the network reconstructs the streamwise velocity fluctuation $u'(x, y, t)$ subject to a no-slip boundary condition at the bottom wall ($y = 0$). REPA-P attaches lightweight $1\times1$ projection heads to selected intermediate layers (encoder blocks, bottleneck, or decoder blocks for U-Net; early, mid-

dle, or late blocks for DiT). These heads map intermediate features to the physical quantity space by first projecting to a hidden dimension (128 for the charge problem) and then to the target channel dimension. We compute PDE and boundary condition residuals on these intermediate predictions and backpropagate them as alignment signals. The mid-layer alignment loss is weighted by $c_{\mathrm{mid}}$ and combined with the standard diffusion data term and output-level physics term when applicable. Following (Bastek et al., 2025), we use $x_0$-prediction with time-dependent residual weighting $\sigma^2(t)$ (see Appendix B and C for the Bayesian derivation). Adaptive temporal modulation strategies similarly exploit stage-specific supervisory signals (Chen et al., 2025c). We train Darcy flow for 120,000 iterations, topology optimization for 150,000 iterations, and the charge potential problem for 120,000 iterations. The varying training budgets reflect the different convergence characteristics of each task. For the charge problem, we use 100 diffusion timesteps with Adam ($\mathrm{lr} = 10^{-4}$) while maintaining an exponential moving average of parameters (decay 0.99). Inference remains unchanged from the baseline, as the projection heads are discarded after training.

**Metrics.** We report physics residual MAE ($R_{\mathrm{MAE}}$) measuring the mean absolute error of PDE and boundary condition residuals on generated samples. For Darcy flow, we additionally report test data MSE on $(K, p)$ reconstructions

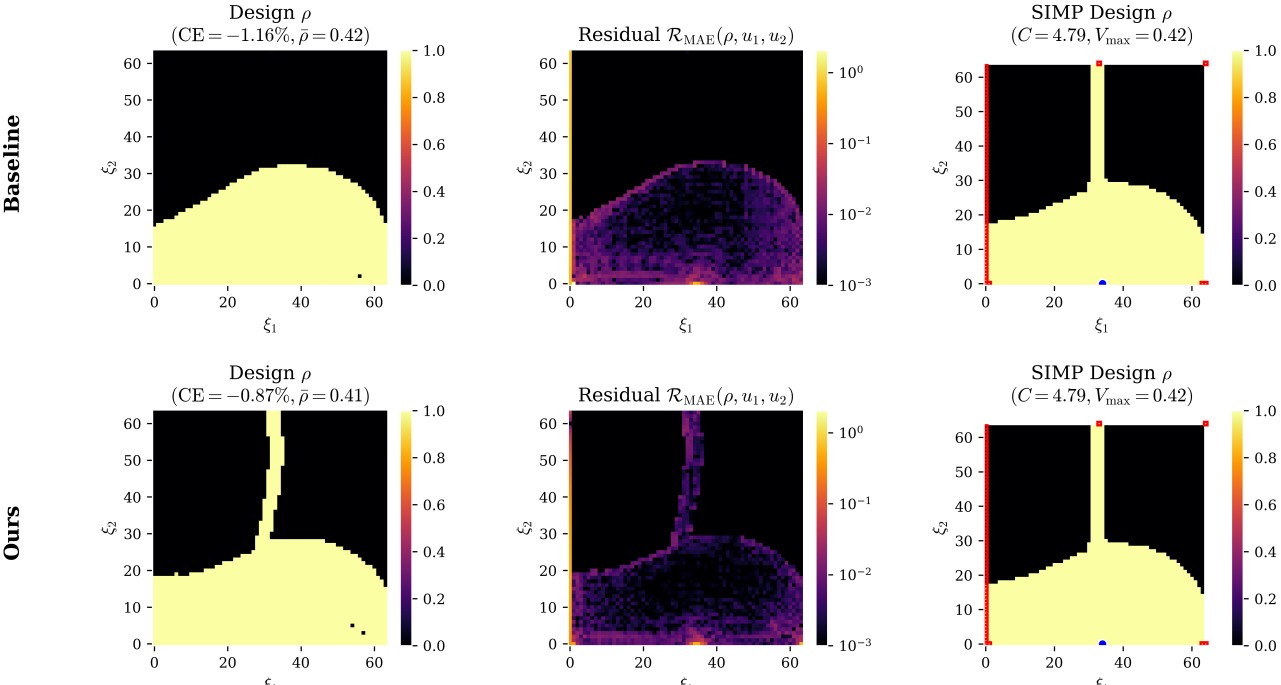

*Figure 3.* **Mechanics topology optimization (Baseline vs. REPA-P).** Each row shows (left→right) the generated density $\rho$ with CE (%) and mean $\bar{\rho}$, the equilibrium residual $\mathcal{R}_{\mathrm{MAE}}(\rho, u_1, u_2)$ (log scale; lower is better), and the SIMP reference with compliance $C$ and volume limit $V_{\max}$ (red: displacement BCs; blue: load). REPA-P yields cleaner slender members and lower residuals than the baseline under the same volume constraint.

and PSNR on the pressure field $p$ for the conditional reconstruction task. For topology optimization, we report compliance error (CE%) measuring how far the generated structure deviates from optimal compliance, and volume fraction error (VFE%) measuring deviation from the target volume constraint. For the charge potential problem, we report the physics loss as the mean absolute residual $\mathcal{L}_{\mathrm{phys}} = \mathrm{mean}(|r|)$ where $r = (-\Delta_h U) - \rho$ is the discrete Poisson residual. For turbulent channel flow, we report PSNR and the physics residual computed via boundary and smoothing regularizers.

### 4.2. Main Results and Analysis

Table 1 reports the performance of REPA-P with the U-Net architecture against four baselines (PG-Diffusion, DiffusionPDE, CoCoGen, and PIDM). REPA-P consistently outperforms all baselines on both data fidelity and physics consistency metrics across every benchmark. Table 2 shows that these gains transfer to the DiT architecture: with comparable parameter counts, intermediate alignment reduces Darcy flow generation data loss by $57.64\%$ and physics loss by $25.73\%$ relative to the PIDM baseline. On topology optimization, REPA-P reduces compliance error under both in-distribution and out-of-distribution boundary conditions regardless of the chosen backbone.

**Darcy Flow.** For unconditional generation, REPA-P reduces both $R_{\mathrm{MAE}}$ and test MSE on the held-out set relative to the PIDM baseline. The generated pairs $(K, p)$ exhibit better adherence to the Darcy PDE while maintaining fidelity to the training distribution. Figure 2 shows representative samples, with REPA-P achieving visibly lower residual magnitudes across the domain. Training converges faster and more stably (Figure 4), and generalization under shifted boundary conditions improves. Inference cost is identical to the standard model since the projection heads are discarded after training.

For conditional generation with sparse reconstruction, we reveal 30% of the target pressure field $p^\star$ as observations via a binary mask $\mathbf{M} \in \{0,1\}^{n \times n}$ with $\frac{1}{n^2} \sum \mathbf{M} = 0.3$. Training augments the diffusion loss with supervision on observed entries and REPA-P alignment on intermediate layers; at inference, observed entries are clamped at each denoising step. REPA-P lowers the masked $\ell_2$ error on unobserved entries and decreases $R_{\mathrm{MAE}}$, with gains persisting across different observation ratios. This demonstrates that intermediate alignment yields more physically consistent reconstructions without extra test-time optimization.

**Topology Optimization.** Table 3 reports performance under both seen and unseen boundary conditions. REPA-P consistently achieves the lowest physics residual ($R_{\mathrm{MAE}}$) and com-

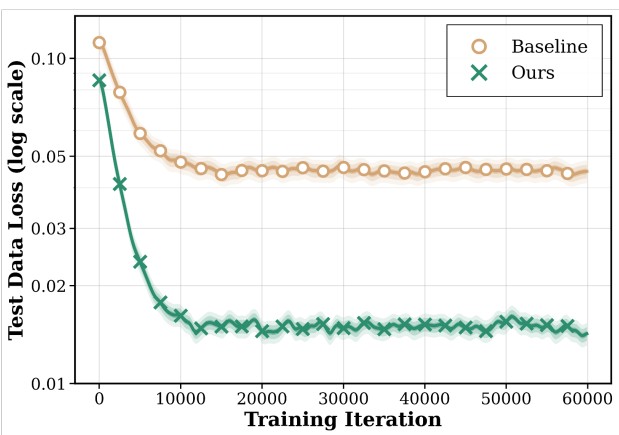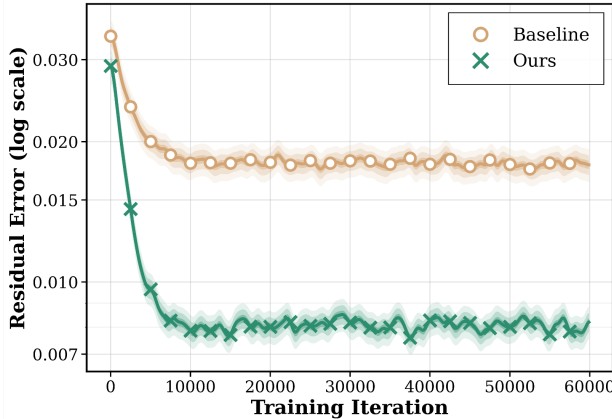

*Figure 4.* Training convergence curves on Darcy flow (first 60K of 120K total iterations). **Left:** Test data loss (log scale). **Right:** Physics residual error (log scale). REPA-P achieves significantly faster convergence and lower final loss on both metrics compared to the baseline. The shaded regions indicate standard deviation across 3 runs. Best viewed in color.

pliance error (CE%), demonstrating that mid-layer alignment improves both physical consistency and task-specific performance. Figure 3 provides qualitative examples, showing that REPA-P produces structures with lower compliance error and reduced physics violations. On the in-distribution test set, REPA-P reduces compliance error substantially; on the challenging out-of-distribution set with unseen boundary conditions, gains persist. These results indicate that mid-layer alignment encourages the network to internalize physical constraints rather than memorizing task-specific patterns, enabling better transfer to novel boundary conditions.

**Turbulent Channel Flow.** To evaluate REPA-P on a more complex fluid dynamics scenario, we introduce a turbulent channel flow benchmark. The task is to reconstruct a DNS $128 \times 48$ $x$-$y$ slice of the streamwise velocity fluctuation $u'(x, y, t)$. Physics consistency is enforced through a no-slip boundary condition at the bottom wall ($u'(x, 0, t) = 0$) and an interior Laplacian regularizer. As shown in Tables 1 and 2, REPA-P significantly enhances both reconstruction quality and physical consistency. Using the U-Net backbone, REPA-P (bottleneck alignment) improves PSNR from 37.64 dB to 39.95 dB while simultaneously reducing the physics residual. The improvements seamlessly translate to the DiT architecture, where REPA-P (middle alignment) boosts PSNR from 38.16 dB to 39.65 dB and lowers the physics residual by over 25%. These results demonstrate that intermediate physical supervision remains effective and robust even on substantially more complex turbulent-flow benchmarks.

**Electrostatic Charge Potential.** On the electrostatic charge potential problem, where the model generates the potential field $U$ conditioned on the charge density $\rho$, REPA-P reduces physics loss by up to 66.4% relative to the PIDM baseline. This conditional generation task tests whether

intermediate alignment improves the physical consistency of predicted solutions to the Poisson equation. The result validates our core hypothesis: enforcing physical decodability at intermediate layers provides meaningful gradient signals that complement output-level constraints, even when the source term is given as conditioning input. Additional qualitative results for all tasks are provided in Appendix E.

### 4.3. Ablation Study

We conduct ablation studies to investigate the impact of alignment position on model performance. Table 4 presents results for Darcy flow, Table 3 shows topology optimization results, and Table 5 presents detailed results for the charge potential problem.

**Impact of Alignment Position.** We systematically evaluate placing projection heads at different positions within the U-Net architecture: encoder blocks, bottleneck, decoder blocks, and output layer. To isolate the effect of alignment position, we follow a two-stage experimental procedure: (1) we first compare different alignment positions using fixed default hyperparameters ($c_{\mathrm{mid}} = 0.01$, projection head hidden dimension 128), and (2) we then perform hyperparameter tuning on the best-performing position identified in stage (1). This design allows us to fairly assess the impact of alignment position independently of task-specific hyperparameter optimization.

For Darcy flow (Table 4), stage (1) position ablations show that bottleneck alignment achieves the best data loss and physics consistency for unconditional generation, while output-level alignment performs best on reconstruction physics, indicating complementary strengths across positions. We therefore choose the bottleneck and tune $c_{\mathrm{mid}}$ (stage 2), obtaining REPA-P ($c_{\mathrm{mid}} = 0.1$) with the best overall trade-off.

*Table 1.* Summary of main results across all benchmarks using the **U-Net** architecture, comparing multiple baselines against our proposed REPA-P. We report key metrics for each task: Data/Phys. loss for Darcy generation, PSNR/Phys. for reconstruction, CE%/Phys. for topology optimization, Data/Phys. for charge, and PSNR/Phys. for turbulence. Best results in **bold**. ↓: lower is better; ↑: higher is better.

| | Darcy Flow | | | | Topology Optimization | | | | Charge | | Turbulence | |
| | Generation | | Reconstruction | | In-Distribution | | Out-of-Distribution | | Generation | | Reconstruction | |
| Method | Data↓ | Phys.↓ | PSNR↑ | Phys.↓ | CE%↓ | Phys.↓ | CE%↓ | Phys.↓ | Data↓ | Phys.↓ | PSNR↑ | Phys.↓ |
|---|---|---|---|---|---|---|---|---|---|---|---|---|
| PG-Diffusion | 0.0973 | 0.1041 | 35.89 | 0.1157 | 15.57 | 7.9e-3 | 13.45 | 7.6e-3 | 0.1055 | 0.868 | 37.68 | 2.15e-3 |
| REPA-P (ours) | **0.0431** | **0.0734** | **37.24** | **0.065** | **7.49** | **5.1e-3** | **8.21** | **6.3e-3** | **0.0543** | **0.344** | **38.97** | **1.41e-3** |
| *Rel. Improv.* | *55.7%* | *29.5%* | *+1.35dB* | *43.6%* | *51.9%* | *35.4%* | *39.0%* | *17.1%* | *48.5%* | *60.4%* | *+1.29dB* | *34.4%* |
| DiffusionPDE | 0.0879 | 0.1136 | 36.49 | 0.0973 | 17.62 | 9.4e-3 | 19.58 | 9.7e-3 | 0.1243 | 0.966 | 38.33 | 1.73e-3 |
| REPA-P (ours) | **0.0342** | **0.0678** | **37.98** | **0.0611** | **8.40** | **6.2e-3** | **9.93** | **7.2e-3** | **0.0635** | **0.454** | **39.45** | **1.24e-3** |
| *Rel. Improv.* | *61.1%* | *40.3%* | *+1.49dB* | *37.2%* | *52.3%* | *34.0%* | *49.3%* | *25.8%* | *48.9%* | *53.0%* | *+1.12dB* | *28.3%* |
| CoCoGen | 0.1231 | 0.1134 | 34.47 | 0.1047 | 20.16 | 2.3e-2 | 17.34 | 1.9e-2 | 0.1967 | 1.327 | 38.40 | 2.30e-3 |
| REPA-P (ours) | **0.0921** | **0.0804** | **37.05** | **0.0833** | **11.26** | **8.4e-3** | **10.82** | **8.1e-3** | **0.1138** | **0.550** | **39.79** | **1.59e-3** |
| *Rel. Improv.* | *25.2%* | *29.1%* | *+2.58dB* | *20.4%* | *44.1%* | *63.5%* | *37.6%* | *57.4%* | *42.1%* | *58.6%* | *+1.39dB* | *30.9%* |
| PIDM | 0.0180 | 0.0260 | 36.23 | 0.0234 | 9.24 | 5.2e-3 | 7.93 | 5.1e-3 | 0.0168 | 0.381 | 37.64 | 1.91e-3 |
| REPA-P (ours) | **0.0119** | **0.0143** | **38.41** | **0.0142** | **4.17** | **4.5e-3** | **5.05** | **4.9e-3** | **0.0081** | **0.128** | **39.95** | **1.75e-3** |
| *Rel. Improv.* | *33.9%* | *45.0%* | *+2.18dB* | *39.3%* | *54.9%* | *13.5%* | *36.3%* | *3.9%* | *51.8%* | *66.4%* | *+2.31dB* | *8.4%* |

*Table 2.* Performance comparison on the **DiT** architecture (PIDM baseline vs. REPA-P). We report data fitting and physics consistency metrics across all benchmarks. REPA-P provides consistent performance gains when transitioning from U-Net to DiT. Best results in **bold**. ↓: lower is better; ↑: higher is better.

| | Darcy Flow | | | | Topology Optimization | | | | Charge | | Turbulence | |
| | Generation | | Reconstruction | | In-Distribution | | Out-of-Distribution | | Generation | | Reconstruction | |
| Method | Data↓ | Phys.↓ | PSNR↑ | Phys.↓ | CE%↓ | Phys.↓ | CE%↓ | Phys.↓ | Data↓ | Phys.↓ | PSNR↑ | Phys.↓ |
|---|---|---|---|---|---|---|---|---|---|---|---|---|
| PIDM | 0.0831 | 0.0719 | 35.34 | 0.0692 | 14.93 | 1.38e-3 | 10.11 | 1.42e-3 | 0.0135 | 0.367 | 38.16 | 1.81e-3 |
| REPA-P (ours) | **0.0352** | **0.0534** | **37.39** | **0.0378** | **6.79** | **1.31e-3** | **5.32** | **1.30e-3** | **0.0093** | **0.220** | **39.65** | **1.35e-3** |
| *Rel. Improv.* | *57.64%* | *25.73%* | *+2.05dB* | *45.4%* | *54.5%* | *5.1%* | *47.4%* | *8.5%* | *31.1%* | *40.0%* | *+1.49dB* | *25.4%* |

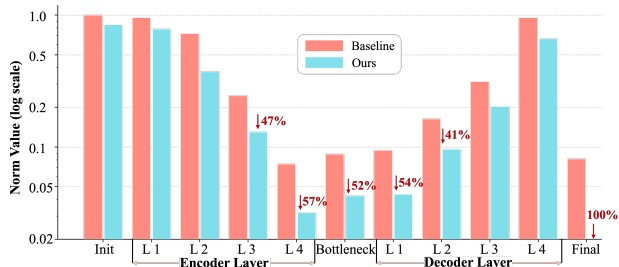

*Figure 5.* Physics residual (normalized, log scale) across U-Net layers. **Baseline** (red) applies physics loss only at output; **Ours** (blue) applies REPA-P alignment at intermediate layers, achieving 47%-100% reduction.

Table 5 details the ablation on the charge potential problem. Bottleneck alignment yields the best performance, reducing data loss by 51.8% and physics loss by 66.4% compared to the baseline. Encoder and decoder alignments also pro-

vide substantial improvements, consistently outperforming output-only alignment. This confirms that mid-layer supervision effectively captures essential physical relationships in the compressed bottleneck representation. Hyperparameter sensitivity is analyzed in the following section.

**Cross-task Consistency.** Across all tasks, mid-layer alignment consistently outperforms output-only alignment and baselines. Figure 5 confirms that REPA-P enforces physical decodability at intermediate layers, reducing residuals by 47%–100% and preventing deferred physics reasoning. While different positions offer complementary strengths (Tables 4 and 3; DiT position ablations in Appendix F.3), optimized bottleneck alignment achieves the best overall trade-off between expressiveness and physical interpretability. These results validate our core hypothesis: intermediate physical decoding creates short-path gradient signals that break shortcut learning (see Appendix D for a formal gradient flow analysis).

*Table 3.* Topology optimization results under in-distribution (ID) and out-of-distribution (OOD) boundary conditions. **Bold**: best results among single-position ablations; underline: second-best. ↓: lower is better.

| Method | In-Distribution | | | Out-of-Distribution | | |
|---|---|---|---|---|---|---|
| | Phys. | CE% | VFE% | Phys. | CE% | VFE% |
| Baseline | 5.2e-3 | 9.24 | 3.38 | 5.1e-3 | 7.93 | 3.20 |
| + Encoder | 4.5e-3 | 4.17 | 3.02 | 5.3e-3 | 9.07 | 3.02 |
| + Bottleneck | 5.3e-3 | 7.21 | 3.25 | 4.9e-3 | 5.05 | 3.22 |
| + Decoder | 6.7e-3 | 8.67 | 3.64 | 6.2e-3 | 10.02 | 3.47 |
| + Output | 5.5e-3 | 7.47 | 3.96 | 5.0e-3 | 7.98 | 3.42 |
| REPA-P (ours) | **4.5e-3** | **4.17** | **3.02** | **4.9e-3** | **5.05** | 3.02 |

*Table 4.* Darcy Flow results comparing different REPA-P alignment positions. **Bold**: best results among single-position ablations; underline: second-best. ↓: lower is better; ↑: higher is better.

| Method | Generation | | Reconstruction | |
|---|---|---|---|---|
| | Data↓ | Phys.↓ | PSNR↑ | Phys.↓ |
| Baseline | 0.0180 | 0.0260 | 36.23 | 0.0234 |
| + Encoder | 0.0133 | 0.0194 | 37.22 | 0.0201 |
| + Bottleneck | 0.0119 | 0.0143 | 38.41 | 0.0173 |
| + Decoder | 0.0126 | 0.0158 | 38.01 | 0.0182 |
| + Output | 0.0162 | 0.0194 | 38.21 | 0.0142 |
| REPA-P (ours) | **0.0119** | **0.0143** | **38.41** | **0.0142** |

**Hyperparameter Sensitivity.** Following the selection of bottleneck alignment in stage (1), we investigate the sensitivity of REPA-P to key hyperparameters in stage (2), including the physics loss weight $c_{\mathrm{mid}}$ and projection head hidden dimension. REPA-P demonstrates robustness across a wide range, with optimal $c_{\mathrm{mid}}$ spanning 0.005–0.1 across tasks and all tested head dimensions (32–256) substantially outperforming the baseline (see Appendix F.1–F.2 for per-task sensitivity).

## 5. Conclusion

In this paper, we address the critical issue of shortcut learning in scientific diffusion models, where networks memorize surface statistics rather than internalizing governing laws. We propose **REPA-P**, a representation alignment framework that enforces physical decodability directly within the model's intermediate layers. By supervising latent features with first-principles PDE residuals, REPA-P compels the network to "think in physics," effectively breaking these spurious correlations. Our experiments across four PDE benchmarks demonstrate that this internal alignment significantly accelerates convergence and enhances out-of-distribution robustness without inference overhead, advancing toward scientifically trustworthy generative models whose internal representations reflect governing physical laws.

*Table 5.* Ablation study on the electrostatic charge potential problem: data loss (L2 error) and physics loss (residual) across different alignment positions. **Bold**: best results among single-position ablations; underline: second-best. ↓: lower is better.

| Method | Data Loss↓ | Phys.↓ |
|---|---|---|
| Baseline | 1.680e-2 | 0.381 |
| + Encoder | 9.944e-3 | 0.186 |
| + Bottleneck | 8.099e-3 | 0.128 |
| + Decoder | 9.802e-3 | 0.185 |
| + Output | 1.041e-2 | 0.188 |
| REPA-P (ours) | **8.099e-3** | **0.128** |

## Acknowledgements

This work was supported by the Guangdong Basic and Applied Basic Research Foundation (Grant No. 2026A1515011579), the HKUST-HKUST(GZ) 1+1+1 Joint Funding Program (Grant No. C_2025_031), and the Guangzhou-HKUST(GZ) Joint Funding Program (Grant No. 2023A03J0008), Education Bureau of Guangzhou Municipality. This work was also supported by Jiangsu Industrial Technology Research Institute (JITRI) and Wuxi National High-Tech District (WND).

## Impact Statement

This paper advances machine learning methods for AI for Science. We propose REPA-P to improve the physical reliability and out-of-distribution robustness of generative models used in scientific simulation and engineering design (e.g., fluid dynamics and structural topology optimization) by mitigating shortcut learning and encouraging physics-consistent internal representations. If adopted, the method may enable more trustworthy and efficient surrogate modeling pipelines, potentially benefiting downstream scientific discovery and industrial design. REPA-P assumes access to known governing equations and differentiable residual operators, and it introduces additional residual evaluations during training, which may increase computational cost. Our empirical validation is currently limited to the studied benchmarks and discretizations. We plan to extend the approach to more complex and practical settings, such as 3D problems, unstructured meshes, coupled multi-physics systems, and larger-scale engineering workloads, as well as to investigate ways to reduce the training overhead and handle partially unknown or noisy physical constraints.

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

# A. Experimental Details

This section provides additional details on the experimental setup, residual computation, and evaluation metrics for each benchmark task.

## A.1. Darcy Flow

**Problem Formulation.** We study steady two-dimensional Darcy flow governed by the elliptic PDE

$$-\nabla \cdot (K(\xi)\nabla p(\xi)) = f_s(\xi), \quad \xi \in \Omega = [0,1]^2, \tag{A.1}$$

where $K(\xi) > 0$ is the permeability field, $p(\xi)$ is the pressure field, and $f_s(\xi)$ is a source term. We impose homogeneous Neumann boundary conditions $\partial p/\partial n = 0$ on $\partial\Omega$. Since the PDE depends only on pressure gradients, the solution is determined up to an additive constant; we enforce a zero-mean constraint on $p$ for numerical stability (see Eq. 8 in the main text).

**Data Generation.** The permeability field $K(\xi)$ is sampled from a log-Gaussian random field with Matérn covariance kernel (Zhu & Zabaras, 2018). Specifically, we draw $\log K \sim \mathcal{GP}(0, k_\nu)$ where $k_\nu$ is the Matérn kernel with smoothness parameter $\nu = 2.5$ and length scale $\ell = 0.1$. The pressure field $p(\xi)$ is then obtained by solving (A.1) numerically on a $64 \times 64$ uniform grid using second-order central finite differences. This yields paired samples $(K, p) \in \mathbb{R}^{64 \times 64 \times 2}$.

**Residual Computation.** The discrete PDE residual is computed using second-order central finite differences. Expanding the divergence operator $-\nabla \cdot (K\nabla p)$ via the product rule yields

$$-\nabla \cdot (K\nabla p) = -K\Delta p - \nabla K \cdot \nabla p, \tag{A.2}$$

where $\Delta p = \partial_{\xi_1}^2 p + \partial_{\xi_2}^2 p$ is the Laplacian. For interior grid points $(i,j)$ with $1 \le i, j \le n-2$ (where $n = 64$), we discretize each term using second-order central differences:

$$\begin{aligned} R_{\text{PDE}}[i,j] = & -K_{i,j}\left(\frac{p_{i+1,j} - 2p_{i,j} + p_{i-1,j}}{h^2} + \frac{p_{i,j+1} - 2p_{i,j} + p_{i,j-1}}{h^2}\right) \\ & -\frac{K_{i+1,j} - K_{i-1,j}}{2h} \cdot \frac{p_{i+1,j} - p_{i-1,j}}{2h} \\ & -\frac{K_{i,j+1} - K_{i,j-1}}{2h} \cdot \frac{p_{i,j+1} - p_{i,j-1}}{2h} - f_s[i,j], \end{aligned} \tag{A.3}$$

where $h = 1/(n-1)$ is the grid spacing.

For homogeneous Neumann boundary conditions $\partial p/\partial n = 0$, the residual on each boundary enforces vanishing normal derivatives:

$$\begin{aligned} R_{\text{BC}}^{\text{top}}[i] = \frac{p_{i,0} - p_{i,1}}{h}, \quad R_{\text{BC}}^{\text{bottom}}[i] = \frac{p_{i,n-1} - p_{i,n-2}}{h}, \\ R_{\text{BC}}^{\text{left}}[j] = \frac{p_{0,j} - p_{1,j}}{h}, \quad R_{\text{BC}}^{\text{right}}[j] = \frac{p_{n-1,j} - p_{n-2,j}}{h}. \end{aligned} \tag{A.4}$$

The total residual vector is $R(K,p) = [R_{\text{PDE}}; R_{\text{BC}}^{\text{top}}; R_{\text{BC}}^{\text{bottom}}; R_{\text{BC}}^{\text{left}}; R_{\text{BC}}^{\text{right}}] \in \mathbb{R}^{d_r}$, where $d_r = (n-2)^2 + 4n$.

**Evaluation Metrics.** The physics residual MAE ($R_{\text{MAE}}$) measures the mean absolute error of the PDE and boundary condition residuals:

$$R_{\text{MAE}} = \frac{1}{N_{\text{test}}} \sum_{i=1}^{N_{\text{test}}} \frac{1}{d_r} \|R(K^{(i)}, p^{(i)})\|_1. \tag{A.5}$$

The data loss measures the mean squared error between generated and ground-truth fields:

$$\text{Data Loss} = \frac{1}{N_{\text{test}}} \sum_{i=1}^{N_{\text{test}}} \left(\|K^{(i)} - \hat{K}^{(i)}\|_2^2 + \|p^{(i)} - \hat{p}^{(i)}\|_2^2\right). \tag{A.6}$$

For conditional reconstruction, we additionally report PSNR on the pressure field:

$$\text{PSNR} = 10 \log_{10} \left( \frac{\max(p)^2}{\text{MSE}(p, \hat{p})} \right), \tag{A.7}$$

and the masked $\ell_2$ error on unobserved entries: $\ell_2^{\text{masked}} = \|(1 - \mathbf{M}) \odot (p - \hat{p})\|_2 / \|1 - \mathbf{M}\|_0$, where $\odot$ denotes element-wise multiplication.

**Training Details.** We use a U-Net backbone with 4 encoder blocks and 4 decoder blocks, each containing two residual blocks with group normalization and SiLU activations. The base channel dimension is 32, doubling at each downsampling stage. Skip connections link corresponding encoder and decoder blocks. The diffusion process uses $T = 1000$ timesteps with a cosine noise schedule. We train for 120,000 iterations using Adam with learning rate $10^{-4}$ and batch size 32.

For REPA-P, we attach $1 \times 1$ convolutional projection heads to the bottleneck and selected decoder blocks. Each head consists of a $1 \times 1$ convolution mapping from the hidden dimension to 2 output channels (for $K$ and $p$), followed by bilinear upsampling to the target resolution $64 \times 64$. The mid-layer alignment weight is set to $c_{\text{mid}} = 0.1$ for the main results, which achieves optimal performance. Ablation studies (Section F.1) explore the sensitivity to this hyperparameter across different values.

## A.2. Topology Optimization

**Problem Formulation.** Two-dimensional structural topology optimization seeks to find an optimal material distribution $\rho(\xi) \in [0, 1]$ that minimizes compliance (maximizes stiffness) subject to mechanical equilibrium and a volume constraint:

$$\min_{\rho} \quad C(\rho) = \mathbf{f}^\top \mathbf{u}, \quad \text{s.t.} \quad \mathbf{K}(\rho)\mathbf{u} = \mathbf{f}, \quad \int_\Omega \rho \, d\xi \leq V_{\text{target}}, \tag{A.8}$$

where $\mathbf{K}(\rho)$ is the global stiffness matrix assembled from element stiffnesses $\mathbf{k}_e(\rho_e) = \rho_e^p \mathbf{k}_e^0$ (SIMP penalization with $p = 3$), $\mathbf{u}$ is the displacement vector, $\mathbf{f}$ is the external load vector, and $V_{\text{target}}$ is the target volume fraction.

**Data Generation.** We follow the dataset from (Bastek et al., 2025), which contains 30,000 optimized structures (Mazé & Ahmed, 2022) on a $64 \times 64$ grid. Each sample consists of a density field $\rho \in [0, 1]^{64 \times 64}$, boundary condition indicators (fixed supports), and load vectors. The dataset covers diverse boundary conditions and target volume fractions ranging from 0.3 to 0.6. The training set contains 24,000 samples, validation set 3,000 samples, and test sets 1,500 samples each for in-distribution (ID) and out-of-distribution (OOD) boundary conditions.

**Residual Computation.** The physics residual for topology optimization consists of three components. The mechanical equilibrium residual measures violation of the finite element equation:

$$R_{\text{eq}} = \|\mathbf{K}(\rho)\mathbf{u} - \mathbf{f}\|_2. \tag{A.9}$$

The volume constraint residual penalizes deviation from the target volume fraction:

$$R_{\text{vol}} = \max \left( 0, \frac{1}{|\Omega|} \sum_{i,j} \rho_{i,j} - V_{\text{target}} \right). \tag{A.10}$$

The density bound residual ensures $\rho \in [0, 1]$:

$$R_{\text{bound}} = \| \max(0, -\rho)\|_2 + \| \max(0, \rho - 1)\|_2. \tag{A.11}$$

The total physics residual is $R_{\text{MAE}} = R_{\text{eq}} + \lambda_{\text{vol}} R_{\text{vol}} + \lambda_{\text{bound}} R_{\text{bound}}$ with $\lambda_{\text{vol}} = \lambda_{\text{bound}} = 1$.

**Evaluation Metrics.** The compliance error (CE%) measures relative deviation from optimal compliance:

$$\text{CE\%} = \frac{1}{N_{\text{test}}} \sum_{i=1}^{N_{\text{test}}} \frac{|C(\hat{\rho}^{(i)}) - C(\rho_{\text{opt}}^{(i)})|}{C(\rho_{\text{opt}}^{(i)})} \times 100\%. \tag{A.12}$$

The volume fraction error (VFE%) measures deviation from target volume:

$$\text{VFE\%} = \frac{1}{N_{\text{test}}} \sum_{i=1}^{N_{\text{test}}} \left| \frac{1}{|\Omega|} \sum_{i,j} \hat{\rho}_{i,j} - V_{\text{target}} \right| \times 100\%. \tag{A.13}$$

**Conditional Generation.** Topology optimization is formulated as a conditional generation task, where the model generates optimal density fields $\rho$ conditioned on boundary conditions and load configurations. The conditioning information is provided as additional input channels concatenated with the noisy density field during training and inference. Specifically, the 4 input channels consist of: (1) the noisy density field $\rho_t$, (2) x-component of the load vector, (3) y-component of the load vector, and (4) boundary condition indicator (binary mask indicating fixed supports). This channel-wise concatenation allows the U-Net to learn the mapping from boundary conditions and loads to optimal material distributions.

**Training Details.** The U-Net backbone is extended to 4 input channels (density, x-load, y-load, boundary indicator) and 1 output channel (density). We use the same architecture as Darcy flow with base channel dimension 128. The diffusion process uses $T = 1000$ timesteps with a cosine noise schedule. Training runs for 150,000 iterations with Adam, learning rate $5 \times 10^{-5}$, and batch size 32. The longer training budget compared to Darcy flow reflects the increased complexity of topology optimization, which involves multiple interacting constraints (mechanical equilibrium, volume fraction, and density bounds) and a significantly larger network (136M vs 9M parameters).

For REPA-P, projection heads are attached to the bottleneck layer. The heads map bottleneck features (512 channels) to the density field via $1 \times 1$ convolution followed by bilinear upsampling. The mid-layer alignment weight is set to $c_{\text{mid}} = 5 \times 10^{-3}$ (0.005) for the main results, which provides the best balance between compliance error and physics consistency. The output physics weight is $c_{\text{out}} = 10^{-3}$. Ablation studies (Section F.1) demonstrate the sensitivity to different weight values.

## A.3. Electrostatic Charge Potential

**Problem Formulation.** We study a two-dimensional electrostatic charge–potential problem on the square domain $\Omega = [0,1]^2$, represented on a $P \times P$ grid with $P = 64$ including boundary pixels, under homogeneous Dirichlet boundary conditions $U|_{\partial\Omega} = 0$. The governing equation is the Poisson equation in normalized units:

$$(-\Delta)U(\xi) = \rho(\xi), \quad \xi \in \Omega, \quad U|_{\partial\Omega} = 0, \tag{A.14}$$

where $U$ is the electric potential and $\rho$ is the (signed) charge density.

**Discretization and Grid Spacing.** We represent the solution on a full $P \times P$ grid (including boundary pixels) with $P = 64$ and grid spacing $h = 1/(P-1) = 1/63$. The interior has size $N \times N$ with $N = P - 2 = 62$. We use the standard 5-point finite-difference Laplacian on the interior:

$$(\Delta_h U)_{i,j} = \frac{U_{i+1,j} + U_{i-1,j} + U_{i,j+1} + U_{i,j-1} - 4U_{i,j}}{h^2}, \tag{A.15}$$

and define $(-\Delta_h U) = -(\Delta_h U)$.

**Data Generation.** Each sample is generated synthetically by placing $K = 2$ random point charges on the interior grid. Each charge has magnitude $q_k \sim \text{Uniform}(0.5, 1.5)$ with random sign. Each charge is deposited onto the nearest single interior grid node $(i_k, j_k)$, producing a sparse discrete charge density:

$$\rho_{i_k, j_k} \leftarrow \rho_{i_k, j_k} + \frac{q_k}{h^2}, \tag{A.16}$$

with all other entries zero. We then solve the discrete Poisson system $(-\Delta_h)U = \rho$ on interior nodes using a discrete sine transform (DST) based solver (diagonalizing the Laplacian in the sine basis). The interior solution is embedded into a full $64 \times 64$ grid by setting boundary values to zero, yielding paired fields $(\rho, U)$ where $\rho \in \mathbb{R}^{64 \times 64}$ serves as the conditioning input and $U \in \mathbb{R}^{64 \times 64}$ is the target output.

**Residual Computation.** Given the predicted potential $\hat{U}$ and the conditioning charge density $\rho$, the discrete Poisson residual uses the standard 5-point finite difference Laplacian:

$$R[i,j] = (-\Delta_h \hat{U})_{i,j} - \rho_{i,j}, \tag{A.17}$$

for interior points $1 \le i, j \le P - 2$. Boundary residuals enforce homogeneous Dirichlet conditions: $R_{\text{BC}}[i,j] = \hat{U}_{i,j}$ for $(i,j) \in \partial\Omega_h$. Note that the charge density $\rho$ is the conditioning input (not predicted), so the residual measures how well the predicted potential $\hat{U}$ satisfies the Poisson equation for the given source term.

**Evaluation Metric.** The reported physics loss is the mean absolute residual over all tensor entries:

$$\mathcal{L}_{\text{phys}} = \text{mean}(|R|), \tag{A.18}$$

i.e., the mean absolute value of the residual tensor for each sample, averaged over the batch.

**Conditional Generation.** The electrostatic charge potential problem is formulated as a conditional generation task, where the model generates the electric potential field $U$ conditioned on the charge density field $\rho$. This represents a forward problem: given the source term $\rho$ (encoding charge positions and magnitudes), predict the resulting potential $U$ that satisfies the Poisson equation $(-\Delta)U = \rho$. The conditioning information $\rho$ is provided as an additional input channel concatenated with the noisy potential field during training. At inference, the model takes a charge density field $\rho$ as input and generates the corresponding potential $U$ through the reverse diffusion process. The physics constraint is enforced through the residual loss, which ensures that the generated potential $U$ is consistent with the input charge density $\rho$ according to the governing PDE.

**Training Details.** The score network is a U-Net operating on 2-channel $64 \times 64$ inputs (noisy potential $U_t$ concatenated with the conditioning charge density $\rho$) and producing 1-channel output (predicted potential $U$). We use 3 encoder and 3 decoder blocks with base channel dimension 32. The diffusion process uses $T = 100$ timesteps with mean $x_0$ estimation (no DDIM sampling).

Training minimizes a weighted combination of the diffusion data term and a physics-informed virtual likelihood based on the Poisson residual: $\mathcal{L} = c_{\text{data}}\mathcal{L}_{\text{data}} + c_{\text{residual}}\mathcal{L}_{\text{residual}}$ with $c_{\text{data}} = 1$ and $c_{\text{residual}} = 10^{-2}$. The residual loss evaluates $(-\Delta_h \hat{U}) - \rho$ where $\hat{U}$ is the predicted potential and $\rho$ is the conditioning charge density. No gradient guidance or correction steps are used during inference. We train for 120,000 iterations with Adam (lr $= 10^{-4}$), batch size 32, and maintain an EMA of parameters with decay 0.99. We construct training and validation splits with 200,000 and 2,048 samples respectively.

For REPA-P, bottleneck projection heads with hidden dimension 128 are enabled with an additional physics loss weight $c_{\text{projection}} = 10^{-2}$. Each head consists of two $1 \times 1$ convolutions with ReLU activation, mapping to 1 output channel (potential $U$) followed by bilinear upsampling. The physics residual is computed using the predicted potential and the conditioning charge density $\rho$. The projection heads are discarded after training, so inference cost remains unchanged.

### A.4. Turbulent Channel Flow

**Problem Formulation.** We study a turbulent channel flow scenario where the goal is to reconstruct the high-fidelity streamwise velocity fluctuation $u'(x, y, t)$ on a two-dimensional $x$-$y$ slice. Physics consistency is enforced through an interior Laplacian smoothing regularizer to penalize unphysical high-frequency artifacts, alongside a strict no-slip boundary condition at the bottom wall ($y = 0$).

**Data Generation.** The dataset consists of Direct Numerical Simulation (DNS) snapshots of turbulent channel flow. We extract 2D slices of the streamwise velocity fluctuation $u'$ discretized on a $128 \times 48$ grid, which effectively captures both the near-wall steep gradients and the outer large-scale flow structures.

**Residual Computation.** The physics residual consists of two components: a boundary condition penalty and an interior smoothness constraint. For the no-slip boundary condition at the bottom wall ($y = 0$), the residual enforces zero velocity fluctuation:

$$R_{\text{BC}}[i] = \hat{u}'_{i,0}, \quad \text{for } 1 \le i \le 128. \tag{A.19}$$

For the interior domain, we apply a Laplacian operator to encourage spatial smoothness and regularize artificial high-frequency noise inherent in the generative process, yielding the residual:

$$R_{\text{smooth}}[i, j] = (\Delta_h \hat{u}')_{i,j}, \tag{A.20}$$

where $\Delta_h$ is the standard discrete Laplacian. The total physics residual is computed as the weighted sum of the boundary and smoothing residuals.

**Evaluation Metric.** We evaluate the reconstruction fidelity using Peak Signal-to-Noise Ratio (PSNR) against the ground-truth DNS fields. The physical consistency is measured by the total absolute residual of the boundary and smoothing constraints mentioned above.

**Training Details.** The score network utilizes the U-Net and the DiT backbones described in the main text. The model operates on the $128 \times 48$ grid. The diffusion process uses $T = 1000$ timesteps with a cosine noise schedule. For REPA-P, projection heads are attached to the bottleneck layer of the U-Net or the middle blocks of the DiT. The mid-layer alignment weight is set to $c_{\text{mid}} = 0.01$, and the model is trained with Adam optimizer using a learning rate of $10^{-4}$.

### A.5. Network Architecture

Table A.1 summarizes the U-Net architecture details for each benchmark task.

*Table A.1.* U-Net architecture details for each benchmark task.

| Component | Darcy Flow | Topology Opt. | Charge Potential |
|---|---|---|---|
| Input channels | 2 | 4 | 2 $(U_t + \rho)$ |
| Output channels | 2 | 1 | 1 $(U)$ |
| Base channels | 32 | 128 | 32 |
| Channel multipliers | [1,2,4,8] | [1,2,4,8] | [1,2,4] |
| Attention resolutions | [8,16] | [8,16] | [16] |
| Residual blocks per level | 2 | 2 | 2 |
| Dropout | 0.0 | 0.1 | 0.0 |
| Total parameters | 9M | 136M | 6M |

## B. Virtual-Observable Derivation of the Physics Objective

**Augmented likelihood view.** Standard diffusion training maximizes the data likelihood (or an ELBO) for samples $x_0 \sim q(x_0)$. To incorporate physical laws, we introduce a virtual observation stating that the discretized PDE/BC residual should be zero. Let

$$\mathcal{O} := \{\hat{r} = 0\}, \qquad \hat{r} \in \mathbb{R}^{d_r}. \tag{B.21}$$

We model $\hat{r}$ as a noisy measurement of residual validity:

$$p(\hat{r} \mid x_0, t) = \mathcal{N}(\hat{r}; R(x_0), \nu^2(t)I), \tag{B.22}$$

where $R(\cdot)$ stacks interior and boundary residuals, and $\nu^2(t)$ controls constraint tolerance at diffusion timestep $t$.

Conditioning on the event $\mathcal{O}$ yields an augmented (unnormalized) joint objective

$$\log p_\theta(x_0, \mathcal{O}) = \log p_\theta(x_0) + \log p(\hat{r} = 0 \mid x_0, t) + \text{const.} \tag{B.23}$$

Thus, maximizing $\log p(\hat{r} = 0 \mid x_0, t)$ under the model prediction is equivalent to minimizing a residual-weighted negative log-likelihood.

**Lemma B.1** (Gaussian virtual observation yields squared residual loss). *Under* (B.22),

$$-\log p(\hat{r} = 0 \mid x_0, t) = \frac{1}{2\nu^2(t)} \|R(x_0)\|_2^2 + const. \tag{B.24}$$

*Proof.* Substitute $\hat{r} = 0$ into the Gaussian density and expand the quadratic form. $\square$

**From $x_0$ to diffusion training.** In diffusion training, we observe $(x_t, t)$ and predict $\hat{x}_0 = \hat{x}_0(x_t, t; \theta)$. We therefore maximize the physical likelihood under the predicted clean field:

$$\mathcal{L}_{\text{phys}}(t) := -\mathbb{E}_{x_0, \epsilon}\Big[\log p(\hat{r} = 0 \mid \hat{x}_0(x_t, t), t)\Big] \equiv \mathbb{E}_{x_0, \epsilon}\Big[\frac{1}{2\nu^2(t)}\|R(\hat{x}_0)\|_2^2\Big]. \tag{B.25}$$

This directly yields the output-end physics term (up to setting $\nu^2(t) = \sigma^2(t)$).

**Extension to intermediate representations (REPA-P).** For each selected layer $\ell \in \mathcal{P}$, REPA-P forms an aligned field $z_\ell = \Pi_\ell(h_\ell)$ and introduces an additional virtual observation $\hat{r}_\ell = 0$:

$$p(\hat{r}_\ell \mid z_\ell, t) = \mathcal{N}\big(\hat{r}_\ell;\ R(z_\ell),\ \nu^2(t)I\big). \tag{B.26}$$

Assuming conditional independence of virtual observations given their respective fields, the total negative log virtual-likelihood is additive:

$$\mathcal{L}_{\text{phys}}^{\text{out}}(t)\ +\ \frac{1}{|\mathcal{P}|}\sum_{\ell \in \mathcal{P}}\mathcal{L}_{\text{phys}}^{(\ell)}(t), \qquad \mathcal{L}_{\text{phys}}^{(\ell)}(t) = \frac{1}{2\nu^2(t)}\|R(z_\ell)\|_2^2, \tag{B.27}$$

which matches the REPA-P objective after setting $\nu^2(t) = \sigma^2(t)$.

# C. Time-Dependent Weighting $\sigma^2(t)$: Uncertainty Calibration

**Key question.** Why scale the physics penalty by $1/\sigma^2(t)$ (and why choose $\sigma^2(t) = \tilde{\beta}_t$)? We provide two complementary justifications: (i) heteroscedastic maximum-likelihood calibration, and (ii) diffusion-consistent uncertainty scheduling.

## C.1. Heteroscedastic Calibration

The virtual observation model (B.22) is heteroscedastic in time. Maximum likelihood therefore prescribes weighting the squared residual by the inverse noise variance, yielding $\|R(\cdot)\|_2^2/(2\nu^2(t))$ (Lemma B.1). Intuitively, when the field estimate at timestep $t$ is less reliable, the residual observation noise should be larger, making the physics penalty softer.

## C.2. Diffusion-Consistent Choice: $\sigma^2(t) = \tilde{\beta}_t$

At timestep $t$, the network infers $\hat{x}_0$ from noisy $x_t$. The uncertainty of this inference naturally depends on the diffusion schedule. We tie the virtual observation variance to the denoising uncertainty scale.

**Delta-method argument.** Assume the predicted clean field has an estimation error

$$\hat{x}_0(x_t, t) = x_0 + \delta_t, \qquad \mathbb{E}[\delta_t] = 0, \qquad \text{Cov}(\delta_t) \approx \tau^2(t)I. \tag{C.28}$$

Linearizing the residual around $x_0$ gives

$$R(\hat{x}_0) \approx R(x_0) + J_R(x_0)\,\delta_t. \tag{C.29}$$

For (approximately) physical data, $R(x_0) \approx 0$, so the residual behaves like a noisy observation:

$$R(\hat{x}_0) \approx J_R(x_0)\,\delta_t, \qquad \mathbb{E}\big[\|R(\hat{x}_0)\|_2^2\big] \approx \tau^2(t)\,\text{tr}\big(J_R(x_0)J_R(x_0)^\top\big). \tag{C.30}$$

Thus, dividing by $\tau^2(t)$ yields a time-normalized physics signal magnitude. Approximating the (unknown) anisotropic residual covariance by an isotropic scalar, we set $\nu^2(t) \propto \tau^2(t)$.

**Choosing $\tau^2(t)$ from the diffusion schedule.** A diffusion-consistent proxy for denoising uncertainty at step $t$ is the DDPM posterior variance

$$\sigma^2(t) = \tilde{\beta}_t = \frac{1 - \bar{\alpha}_{t-1}}{1 - \bar{\alpha}_t}\beta_t, \tag{C.31}$$

which characterizes the intrinsic uncertainty scale of the reverse transition at timestep $t$. We therefore set the virtual observation tolerance to match this scale, $\nu^2(t) = \sigma^2(t) = \tilde{\beta}_t$, obtaining

$$\mathcal{L}_{\text{phys}}(t) = \frac{1}{2}\frac{\|R(\hat{x}_0)\|_2^2}{\sigma^2(t)}. \tag{C.32}$$

**Interpretation as a noise curriculum.** Since $\tilde{\beta}_t$ is larger at high-noise stages and smaller at low-noise stages, the weighting $1/\sigma^2(t)$ automatically enforces physics weakly when $\hat{x}_0$ is uncertain (early timesteps), and strongly when the denoised estimate is reliable (late timesteps), reducing noisy-gradient interference and improving stability in practice.

## D. Gradient Flow Analysis for REPA-P

**Goal.** We formalize why injecting physics losses at intermediate representations provides a stronger and better-conditioned physical learning signal than output-only constraints.

**Notation.** Fix timestep $t$. Let $h_\ell$ be a hidden representation at layer $\ell$. Write the mapping from $h_\ell$ to the output prediction as

$$\hat{x}_0 = g_\ell(h_\ell), \tag{D.33}$$

where $g_\ell$ denotes the composition of all subsequent backbone layers from $\ell$ to output. REPA-P also forms $z_\ell = \Pi_\ell(h_\ell)$.

Define the (scaled) physics losses:

$$\mathcal{L}_{\text{phys}}^{\text{out}}(t) = \frac{1}{2}\frac{\|R(\hat{x}_0)\|_2^2}{\sigma^2(t)}, \qquad \mathcal{L}_{\text{phys}}^{(\ell)}(t) = \frac{1}{2}\frac{\|R(z_\ell)\|_2^2}{\sigma^2(t)}. \tag{D.34}$$

**A generic identity (Gauss-Newton form).** Let $\phi(x) = \frac{1}{2}\|R(x)\|_2^2$. If $R$ is differentiable,

$$\nabla_x\phi(x) = J_R(x)^\top R(x), \tag{D.35}$$

where $J_R(x)$ is the Jacobian of $R$ at $x$.

**Output-only supervision: deep Jacobian chain.** Using (D.35) and the chain rule,

$$\nabla_{h_\ell}\mathcal{L}_{\text{phys}}^{\text{out}}(t) = \frac{1}{\sigma^2(t)}\,J_{g_\ell}(h_\ell)^\top\,J_R(\hat{x}_0)^\top\,R(\hat{x}_0). \tag{D.36}$$

The term $J_{g_\ell}(h_\ell)$ is the Jacobian of a deep mapping and may become ill-conditioned, attenuating the physical signal before it reaches early representations.

**REPA-P mid-layer supervision: short gradient path.** Similarly, for $z_\ell = \Pi_\ell(h_\ell)$,

$$\nabla_{h_\ell}\mathcal{L}_{\text{phys}}^{(\ell)}(t) = \frac{1}{\sigma^2(t)}\,J_{\Pi_\ell}(h_\ell)^\top\,J_R(z_\ell)^\top\,R(z_\ell). \tag{D.37}$$

Compared with (D.36), the gradient path bypasses the remaining backbone and depends only on the lightweight head $\Pi_\ell$, yielding an in-situ physical learning signal.

**Proposition D.1** (Attenuation bound for credit assignment). *Assume $g_\ell$ is a composition of maps $\{g_k\}_{k=\ell+1}^L$ such that $\|J_{g_k}\| \le L_k$ (any operator norm). Then*

$$\big\|\nabla_{h_\ell}\mathcal{L}_{\text{phys}}^{\text{out}}(t)\big\| \le \frac{1}{\sigma^2(t)}\Big(\prod_{k=\ell+1}^L L_k\Big)\,\big\|J_R(\hat{x}_0)^\top R(\hat{x}_0)\big\|. \tag{D.38}$$

*In contrast,*

$$\big\|\nabla_{h_\ell}\mathcal{L}_{\text{phys}}^{(\ell)}(t)\big\| \le \frac{1}{\sigma^2(t)}\,\|J_{\Pi_\ell}(h_\ell)\|\,\big\|J_R(z_\ell)^\top R(z_\ell)\big\|. \tag{D.39}$$

*Proof.* Take norms on (D.36) and (D.37) and apply submultiplicativity: $\|A^\top b\| \le \|A\|\,\|b\|$. Since $J_{g_\ell} = J_{g_L}\cdots J_{g_{\ell+1}}$, we have $\|J_{g_\ell}\| \le \prod_{k=\ell+1}^L L_k$. $\qquad\square$

**Implication.** Proposition D.1 formalizes that output-only physics gradients can be suppressed by a deep product of Jacobian norms, while REPA-P injects physics gradients directly at intermediate layers via $\Pi_\ell$, substantially shortening the credit assignment path. Because the same $1/\sigma^2(t)$ scaling applies to both (D.36) and (D.37), the key distinction is the presence (or absence) of the deep Jacobian chain.

# E. Additional Visualizations

This section provides additional qualitative visualizations complementing the main experimental results. We present detailed field comparisons and residual error visualizations for the electrostatic charge potential task (Figure E.2) and the Darcy flow sparse reconstruction task (Figure E.3). Furthermore, Figure **??** visualizes the reconstruction results for the complex turbulent channel flow task.

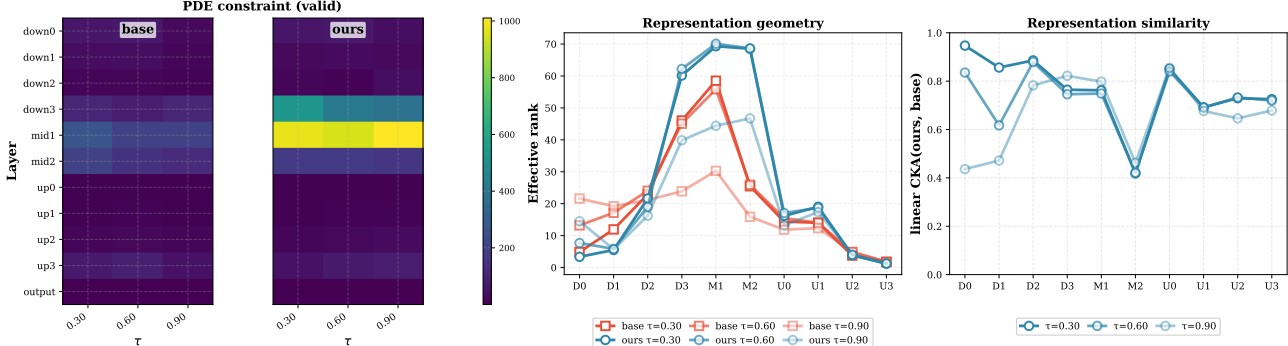

*Figure E.1.* **Representation diagnostics for baseline vs. REPA-P.** *Left:* PDE-constraint responses across network layers under different thresholds $\tau$. Our method shows stronger, more concentrated constraint responses around the bottleneck layers. *Middle:* Representation geometry measured by effective rank. Our model achieves higher rank in the encoder-to-bottleneck region (D0-M2), indicating richer, less-collapsed features. *Right:* Linear CKA similarity between the two models. Similarity remains high for most layers but drops markedly near the bottleneck (M1-M2), suggesting targeted reorganization of the core latent space rather than uniform changes.

# F. Additional Ablation Studies

This section provides additional ablation studies on hyperparameter sensitivity, complementing the alignment position analysis in the main text. We investigate the physics loss weight $c_{\text{mid}}$, the projection head hidden dimension, and the alignment position in the DiT architecture. All projection head experiments are conducted at the optimal alignment position identified in the main text.

## F.1. Effect of Physics Loss Weight

Table F.2 summarizes the sensitivity of REPA-P to the physics loss weight $c_{\text{mid}}$ across all benchmark tasks. The projection head hidden dimension is fixed at 128 for these experiments.

**Observations.** The optimal physics loss weight varies across tasks: Darcy flow benefits from $c_{\text{mid}} = 0.1$, topology optimization from $c_{\text{mid}} = 0.005$, and the charge potential problem from $c_{\text{mid}} = 0.01$. This variation reflects differences in the scale and conditioning of physics residuals across tasks. REPA-P consistently outperforms the baseline across a wide range of weight values (0.001–0.1), demonstrating robustness to this hyperparameter. Very large weights ($c_{\text{mid}} = 0.5$) degrade performance by over-constraining the intermediate representations.

## F.2. Effect of Projection Head Dimension

Table F.3 summarizes the sensitivity of REPA-P to the projection head hidden dimension across all three benchmark tasks. The physics loss weight is fixed at $c_{\text{mid}} = 0.01$ for these experiments.

**Observations.** Projection head dimension shows task-dependent optimal values: Darcy flow and charge potential favor larger dimensions (128–256), while topology optimization performs comparably across dimensions with a slight preference for smaller heads (32–64). This suggests that the required head capacity depends on the complexity of mapping intermediate features to physical quantities. Even dimension 32 provides substantial improvements over the baseline, indicating that the alignment mechanism, rather than head capacity, is the primary driver of performance gains. The lightweight heads (adding $< 1\%$ parameters) make REPA-P practical for large-scale deployment.

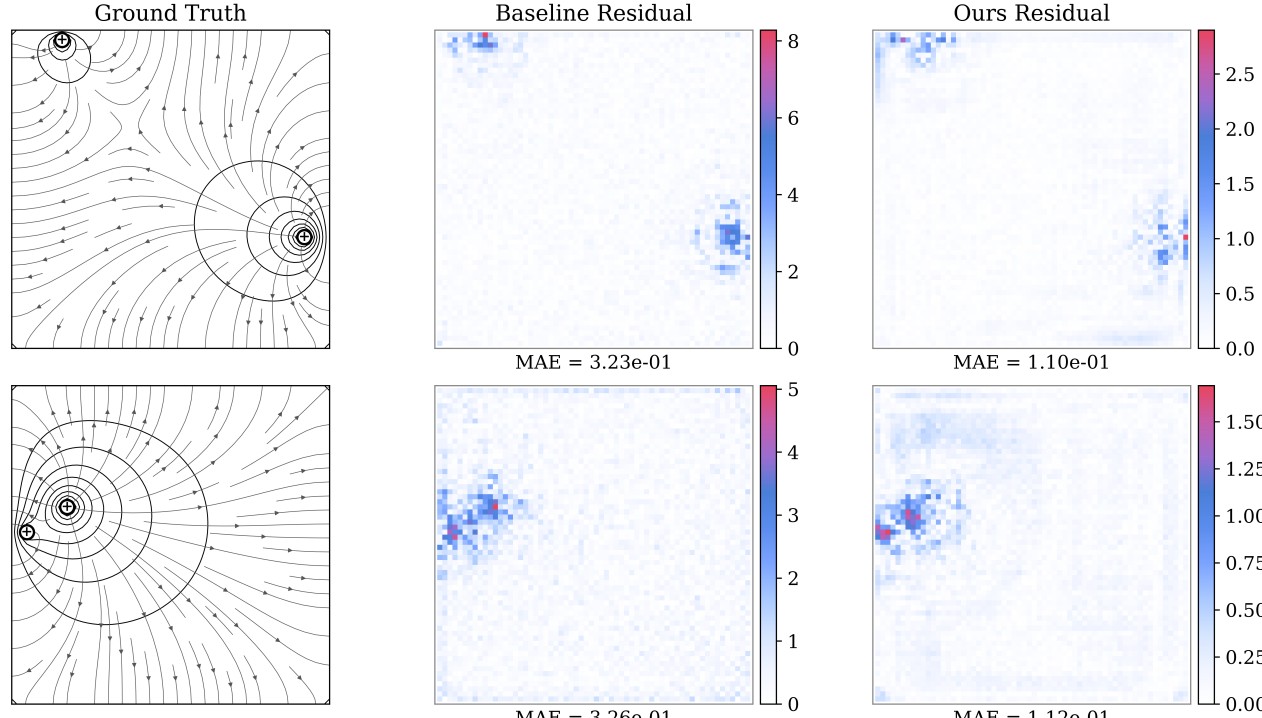

*Figure E.2.* Qualitative comparison on the electrostatic charge potential task. **Left:** Ground truth potential field with charge locations. **Middle:** Absolute PDE residual $|(-\Delta_h U) - \rho|$ for the baseline model; higher values (red) indicate violations of the Poisson equation. **Right:** Absolute PDE residual for REPA-P. REPA-P reduces the residual MAE by $66.4\%$ and achieves more uniform error distribution across the domain, indicating that physics consistency is enforced globally rather than through localized accuracy at the expense of broader violations.

*Table F.2.* Sensitivity analysis of physics loss weight $c_{\text{mid}}$ across all benchmark tasks. Projection head hidden dimension is fixed at 128. Best results (excluding baseline) in **bold**. ↓: lower is better; ↑: higher is better.

| $c_{\text{mid}}$ | Darcy Flow | | Topology Optimization (ID) | | | Charge | Turbulence | |
|---|---|---|---|---|---|---|---|---|
| | Data↓ | Phys.↓ | Phys.↓ | CE%↓ | VFE%↓ | Phys.↓ | PSNR↑ | Phys.↓ |
| Baseline | 0.0180 | 0.0260 | 5.2e-3 | 9.24 | 3.38 | 0.381 | 37.64 | 1.91e-3 |
| 0.001 | 0.0156 | 0.0177 | 7.8e-3 | 11.23 | 4.13 | 0.245 | 38.47 | 1.86e-3 |
| 0.005 | 0.0142 | 0.0165 | **4.5e-3** | **4.17** | **3.02** | 0.189 | 37.91 | 2.04e-3 |
| 0.01 | **0.0119** | 0.0164 | 5.3e-3 | 9.46 | 3.64 | **0.128** | **39.95** | **1.75e-3** |
| 0.05 | 0.0163 | 0.0207 | 6.4e-3 | 6.47 | 4.02 | 0.199 | 38.04 | 1.83e-3 |
| 0.1 | 0.0142 | **0.0143** | 5.8e-3 | 7.84 | 3.38 | 0.254 | 37.85 | 1.81e-3 |
| 0.5 | 0.0152 | 0.0241 | 6.3e-3 | 9.95 | 4.02 | 0.227 | 39.11 | 1.72e-3 |

*Table F.3.* Sensitivity analysis of projection head hidden dimension across all benchmark tasks. Physics loss weight is fixed at $c_{\text{mid}} = 0.01$. Best results (excluding baseline) in **bold**. ↓: lower is better; ↑: higher is better.

| Dim | Darcy Flow | | Topology Optimization (ID) | | | Charge | Turbulence | |
|---|---|---|---|---|---|---|---|---|
| | Data↓ | Phys.↓ | Phys.↓ | CE%↓ | VFE%↓ | Phys.↓ | PSNR↑ | Phys.↓ |
| Baseline | 0.0180 | 0.0260 | 5.2e-3 | 9.24 | 3.38 | 0.381 | 37.64 | 1.91e-3 |
| 256 | 0.0135 | **0.0143** | 6.7e-3 | 6.25 | 4.13 | 0.172 | 38.06 | 1.92e-3 |
| 128 | **0.0119** | 0.0164 | 5.3e-3 | 9.46 | 3.64 | **0.128** | **39.95** | **1.75e-3** |
| 64 | 0.0151 | 0.0178 | **4.5e-3** | 8.37 | **3.18** | 0.216 | 39.57 | 1.77e-3 |
| 32 | 0.0193 | 0.0221 | 5.7e-3 | **4.38** | 3.86 | 0.284 | 38.94 | 1.80e-3 |

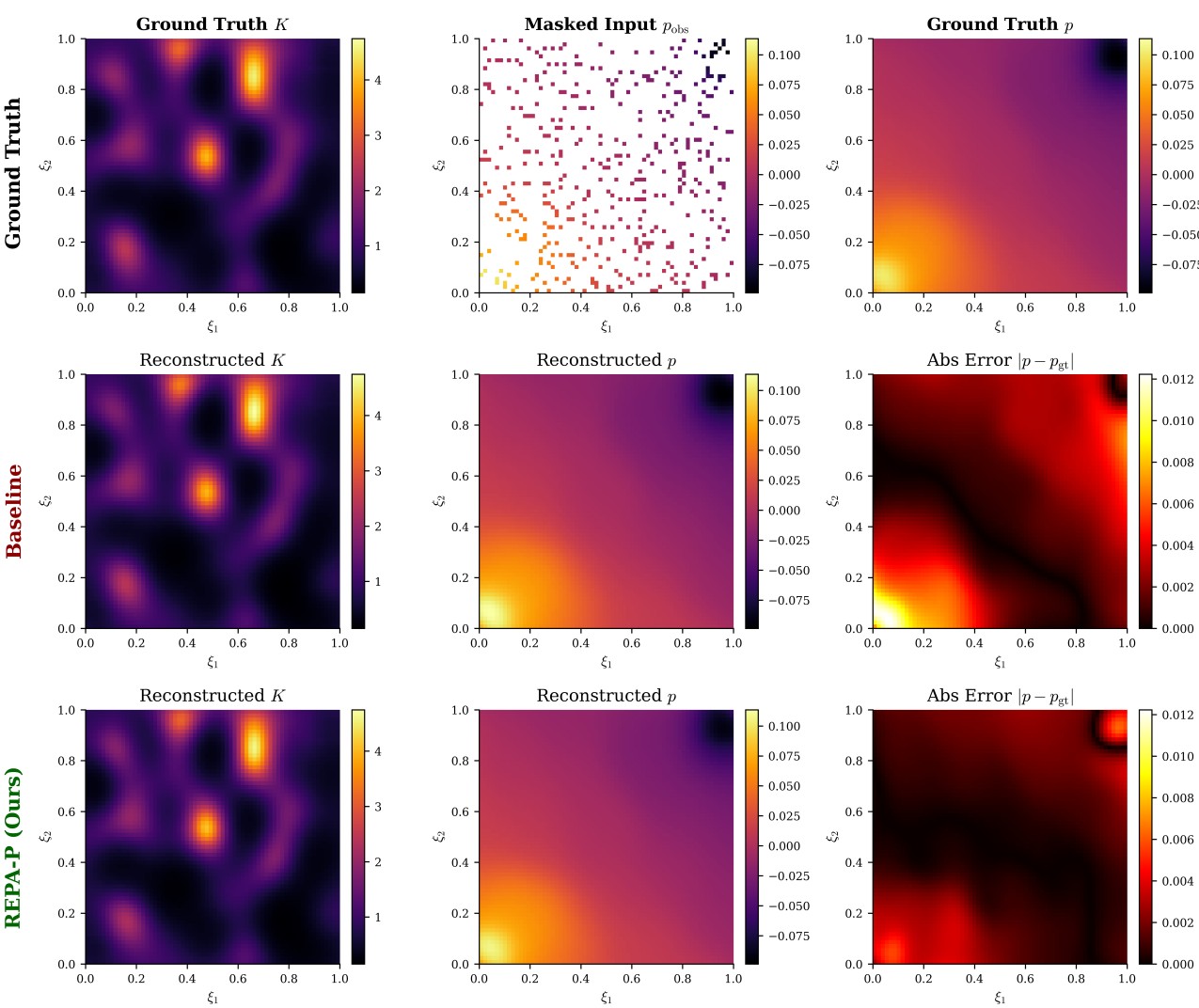

*Figure E.3.* Darcy flow sparse reconstruction from 30% observed pressure measurements. **Top row:** Ground truth. **Middle row:** Baseline. **Bottom row:** REPA-P. Each row shows: (a) permeability $K(\xi)$, (b) pressure $p(\xi)$, and (c) absolute error. REPA-P achieves lower reconstruction error with better preservation of physical structures, demonstrating that intermediate alignment yields more accurate and physically consistent reconstructions.

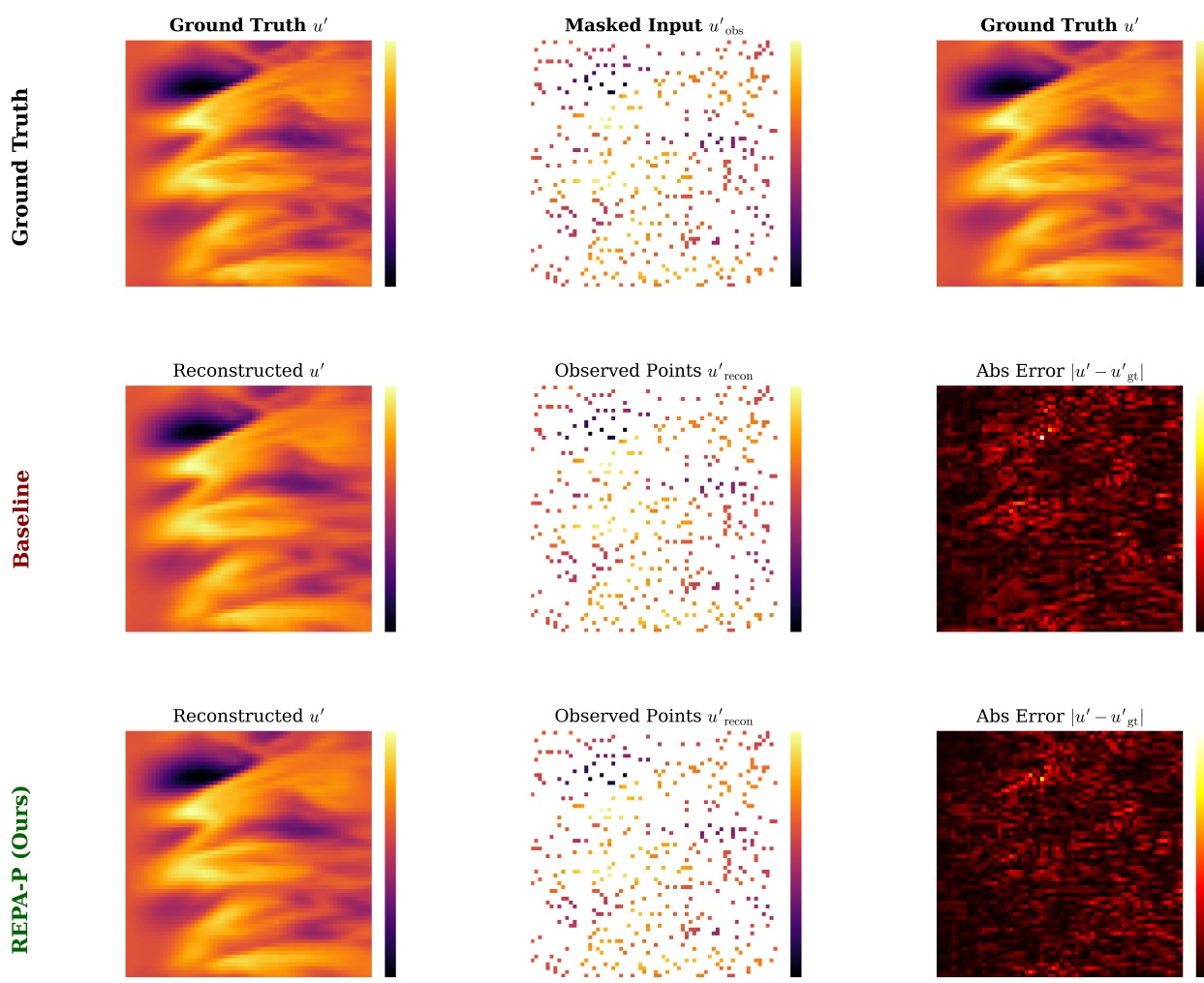

*Figure E.4.* Turbulent channel flow reconstruction from sparse velocity measurements. **Top row:** Ground truth. **Middle row:** Baseline. **Bottom row:** REPA-P. Each row shows: (a) reconstructed streamwise velocity fluctuation $u'$, (b) observed measurements $u'_{obs}$, and (c) absolute error $|u' - u'_{gt}|$. REPA-P achieves lower reconstruction error with better preservation of near-wall flow structures, demonstrating that intermediate alignment yields more accurate and physically consistent reconstructions.

*Table F.4.* Ablation study of REPA-P alignment position within the **DiT architecture** across all benchmarks. We partition the 8-layer DiT into three feature stages and select representative blocks for evaluation. Best results (excluding baseline) in **bold**. ↓: lower is better; ↑: higher is better.

| DiT Position | Darcy Flow | | | | Topology Optimization | | | | Charge | | Turbulence | |
| | Generation | | Reconstruction | | In-Distribution | | Out-of-Distribution | | Generation | | Reconstruction | |
| | Data↓ | Phys.↓ | PSNR↑ | Phys.↓ | CE%↓ | Phys.↓ | CE%↓ | Phys.↓ | Data↓ | Phys.↓ | PSNR↑ | Phys.↓ |
|---|---|---|---|---|---|---|---|---|---|---|---|---|
| Baseline (PIDM) | 0.0831 | 0.0719 | 35.34 | 0.0692 | 14.93 | 1.38e-3 | 10.11 | 1.42e-3 | 0.0135 | 0.367 | 38.16 | 1.81e-3 |
| + Early (Block 2) | 0.0482 | 0.0621 | 36.51 | 0.0512 | 11.54 | 1.42e-3 | 9.35 | 1.36e-3 | 0.0117 | 0.302 | 38.89 | 1.55e-3 |
| + Middle (Block 4) | **0.0352** | **0.0534** | **37.39** | **0.0378** | **6.79** | **1.31e-3** | **5.32** | **1.30e-3** | **0.0093** | **0.220** | **39.65** | **1.35e-3** |
| + Late (Block 6) | 0.0415 | 0.0588 | 36.92 | 0.0455 | 8.43 | 1.33e-3 | 6.46 | 1.32e-3 | 0.0098 | 0.274 | 39.12 | 1.48e-3 |

## F.3. Effect of Alignment Position in DiT Architecture

To verify that the benefits of REPA-P are not restricted to the U-Net topology, we conduct an ablation study on the alignment position within the Diffusion Transformer (DiT) backbone. We partition the 8-layer DiT into three consecutive feature abstraction stages: early (Blocks 1–3), middle (Blocks 4–6), and late (Blocks 7–8). We select the central block of each stage (Block 2, Block 4, and Block 6) as representative positions to attach projection heads, covering the full trajectory of feature evolution from low-level local patterns to high-level semantic representations. Table F.4 presents the complete performance across all four benchmarks.

**Observations.** Consistent with the U-Net results, aligning at the middle stage (Block 4) achieves the optimal trade-off between data fidelity and physical consistency across all tasks. Early-stage alignment fails to decode complex physical relationships due to insufficient feature abstraction, while late-stage alignment suffers from deferred physics reasoning and attenuated gradient signals (see Appendix D). This confirms that mid-layer physical supervision is effective across fundamentally different architectural paradigms (CNN and Transformer).

