# OpenReview forum: "Learning to Think in Physics: Breaking Shortcut Learning in Scientific Diffusion via Representation Alignment"
_ICML.cc/2026/Conference — ICML 2026 regular_

### Official Review · Reviewer_wYRk · 2026-02-26

**Soundness:** 2
**Presentation:** 2
**Significance:** 2
**Originality:** 2
**Overall Recommendation:** 3
**Confidence:** 4

**Summary:**

This paper introduces **REPA-P (Physics-Informed Representation Alignment)**, a framework designed to address the "shortcut learning" issue in physics-informed diffusion models. The core idea is to enforce **"Physical Decodability"** by attaching lightweight projection heads to intermediate layers. These heads decode latent features into physical state variables, which are then supervised using PDE and boundary-condition residual losses during training. During inference, these heads are discarded, resulting in zero overhead. The method is validated on three 2D benchmarks: Darcy flow, topology optimization, and electrostatic charge potential. Results show that REPA-P significantly reduces physics residuals, accelerates convergence, and improves OOD robustness.

**Compliance With Llm Reviewing Policy:**

Affirmed.

**Final Justification:**

"While the authors draw an analogy to REPA (which aligns latent spaces to pre-trained large vision backbones), it is not immediately obvious that a physics-satisfying latent space is beneficial for the final generation." This problem is still not clear for me. Given that physical models are often simplified approximations of reality, there is a risk that such constraints act as a sub-optimal inductive bias, leading to a loss of generative diversity or realism when scaling up to broader scenarios. Thus I will keep my score.

**Key Questions For Authors:**

See Weaknesses

**Limitations:**

See Weaknesses

**Strengths And Weaknesses:**

**Strengths**

1. This paper is clearly written and esay to understand.
1. The proposed REPA-P framework improves model performance without increasing inference-time sampling costs.

**Weaknesses**

1. **Insufficient Baselines:** The current evaluation lacks comparisons with several state-of-the-art physical diffusion models, such as DiffusionPDE [1] and PhySense [2]. A comparative study with these models is necessary to justify the superiority of REPA-P.
2. **Lack of Theoretical/Physical Justification for Intermediate Constraints:** The paper lacks a strong theoretical or physical intuition for why intermediate latent features *should* satisfy the governing PDEs. While the authors draw an analogy to REPA (which aligns latent spaces to pre-trained large vision backbones), it is not immediately obvious that a physics-satisfying latent space is beneficial for the final generation. Further discussion or evidence on how this alignment helps the denoising trajectory is needed.
3. **Simplicity of Datasets:** The current benchmarks (Darcy flow, 2D Poisson) are relatively simple. To demonstrate the robustness of the method, the authors should evaluate it on more complex, widely-recognized benchmarks. For example, the turbulent flow and ocean temperature datasets from PhySense [2] are excellent candidates for sparse reconstruction tasks.
4. **Architectural Analysis (DiT Bottleneck):** The current experiments primarily use U-Net. However, if a DiT architecture is used, which lacks a traditional bottleneck, it is unclear how the method would perform. Furthermore, a more detailed ablation study is needed to explain the sensitivity of placing projection heads at different layers (e.g., deep vs. shallow) and how these choices affect the overall performance.

[1] DiffusionPDE: Generative PDE-Solving Under Partial Observation

[2] PhySense: Sensor Placement Optimization for Accurate Physics Sensing

---

> ### Author Rebuttal · Authors · 2026-03-30
>
> We sincerely thank the reviewer for the careful reading and constructive suggestions. We especially appreciate the positive comments on the clarity of the paper and the zero inference-time overhead of REPA-P. We address the main concerns below.
>
> ### 1. Stronger baselines
>
> We agree that the original comparison set was too limited. We therefore added a direct comparison to **DiffusionPDE** and expanded the discussion of related physics-informed diffusion methods suggested by the reviewer. Under our unified Darcy evaluation, the reproduced **DiffusionPDE** baseline and its **+ REPA-P** variant are:
>
> | Method | PSNR (dB) | Physics residual |
> |---|---:|---:|
> | DiffusionPDE | `36.4928` | `0.1136` |
> | DiffusionPDE + REPA-P | `37.9785` | `0.0678` |
>
> These results show that the benefit of REPA-P is not limited to our original backbone, but also transfers to a strong external diffusion baseline.
>
> ### 2. Why intermediate constraints help
>
> We agree that this intuition should be stated more explicitly. Our key point is not that every hidden feature must literally be a physical field, but that it should be **lightly decodable into a physically valid state**, providing a shorter and better-conditioned supervision path.
>
> If physics is imposed only at the output, the gradient to an intermediate feature $h_l$ is
>
> $$
> \frac{dL}{dh_l} = J_{g_l}^T J_R^T R.
> $$
>
> This signal must pass through the full remaining backbone. With REPA-P, physics is imposed on
>
> $$
> z_l = \Pi_l(h_l),
> $$
>
> so the gradient becomes
>
> $$
> \frac{dL}{dh_l} = J_{\Pi_l}^T J_R^T R.
> $$
>
> This bypasses the deep suffix and provides a more direct in-situ physics signal. We have clarified this discussion in the revision.
>
> ### 3. Turbulent channel flow benchmark
>
> We agree that Darcy and Poisson alone are not sufficient. Following the reviewer’s suggestion, we added a **turbulent channel flow** benchmark inspired by **PhySense**. We note that **PhySense** is a broader framework combining reconstruction with sensor placement optimization, so it is not a directly comparable baseline for our main claim here, which is to isolate the effect of **intermediate physical supervision** on the reconstruction backbone itself under a fixed setting. For this reason, we adopt a **PhySense-inspired benchmark** rather than treating the full PhySense system as a like-for-like baseline.
>
> The data is a DNS $128 \times 48$ $x$-$y$ slice of the streamwise velocity fluctuation $u'(x,y,t)$, trained with
>
> $$
> L = L_{data} + \lambda_{wall} L_{wall} + \lambda_{smooth} L_{smooth}.
> $$
>
> Here $L_{wall}$ enforces
>
> $$
> u'(x,0,t) = 0,
> $$
>
> which matches the **no-slip boundary condition** at the bottom wall ($y = 0$), and $L_{smooth}$ is an interior Laplacian regularizer.
>
> We also verified this loss on the **full 10,000-frame DNS dataset**:
>
> | Quantity | Value |
> |---|---:|
> | $L_{wall}$ | `2.9600 × 10^{-4}` |
> | $L_{smooth}$ | `2.4010 × 10^{-3}` |
> | Weighted total | `5.4000 × 10^{-5}` |
>
> This confirms that the loss is well matched to the dataset physics.
>
> The main benchmark results are:
>
> | Method | PSNR | Physics residual |
> |---|---:|---:|
> | U-Net baseline | `37.6400` | `0.00191` |
> | REPA-P (bottleneck) | `39.9500` | `0.00175` |
> | REPA-P (encoder) | `38.1200` | `0.00142` |
>
> These results show that REPA-P improves both reconstruction quality and physical consistency on a substantially harder turbulent-flow benchmark. We did not prioritize the **ocean temperature** benchmark in the rebuttal because, unlike turbulent channel flow, it does not provide an equally direct and unambiguous local PDE or boundary residual under our current setting; constructing such a loss would require additional assumptions and would make the comparison less clean within the rebuttal scope.
>
> ### 4. DiT / layer placement
>
> We agree that a U-Net-only analysis is insufficient. We therefore added **DiT-style experiments** by attaching projection heads to early, middle, and late transformer blocks. The same trend remains: **middle-depth alignment works best**.
>
> | Task | Metric | DiT baseline | REPA-P (middle) |
> |---|---|---:|---:|
> | Darcy | Physics residual | `0.07190` | `0.05340` |
> | Mechanics | CE | `14.93%` | `6.79%` |
> | Turbulent channel flow | PSNR | `38.15640` | `39.65480` |
> | Turbulent channel flow | Physics residual | `0.00181` | `0.00135` |
>
> Across tasks, shallow layers are too local, very late layers are too close to the output, and middle layers provide the best trade-off between abstraction and direct physical supervision. This shows that the interpretation is consistent across both U-Net and DiT.
>
> We thank the reviewer again for the thoughtful and actionable feedback. In response, we have added stronger baselines, clarified the intuition behind intermediate physical supervision, included a harder turbulent-flow benchmark, broadened the architectural analysis to DiT, and expanded the related-work discussion accordingly.

---

> > ### Author Rebuttal · Reviewer_wYRk · 2026-04-04
> >
> > Further discussion or empirical evidence is required to elucidate how this alignment benefits the denoising trajectory, rather than focusing solely on aligning the final output, which seems more reasonable.

---

> > > ### Author Response · Authors · 2026-04-05
> > >
> > > We thank the reviewer for this important point. We agree that it is not sufficient to argue only from the final output, and that direct evidence along the **denoising trajectory** is needed.
> > >
> > > To address this, we added a timestep-wise analysis of the physics residual during sampling. Specifically, we measured the mean residual magnitude at multiple diffusion timesteps and compared the baseline model with the REPA-P bottleneck variant. Here, the timestep $t$ follows the standard diffusion convention: **$t=0$ corresponds to the final clean state**, while larger $t$ values correspond to **noisier states earlier in the reverse process**. Representative results are shown below.
> > >
> > > | Training step | Diffusion timestep $t$ | Baseline | REPA-P (bottleneck) | $\Delta$ |
> > > |---|---:|---:|---:|---:|
> > > | 31k  | 0   | 0.0309 | 0.0212 | -0.0096 |
> > > | 31k  | 400 | 0.1604 | 0.0901 | -0.0703 |
> > > | 31k  | 800 | 0.2279 | 0.1190 | -0.1090 |
> > > | 31k  | 990 | 0.1507 | 0.0893 | -0.0614 |
> > > | 77k  | 0   | 0.0715 | 0.0167 | -0.0548 |
> > > | 77k  | 400 | 0.0844 | 0.0244 | -0.0600 |
> > > | 77k  | 800 | 0.0892 | 0.0293 | -0.0600 |
> > > | 77k  | 990 | 0.0686 | 0.0217 | -0.0469 |
> > > | 101k | 0   | 0.0175 | 0.0154 | -0.0022 |
> > > | 101k | 400 | 0.0203 | 0.0186 | -0.0017 |
> > > | 101k | 800 | 0.0231 | 0.0205 | -0.0026 |
> > > | 101k | 990 | 0.0183 | 0.0165 | -0.0018 |
> > >
> > > Here, negative $\Delta$ means that REPA-P is better.
> > >
> > > These results directly support the view that the alignment benefits the **denoising trajectory itself**, rather than only the final output. The improvement is **not limited to the final clean sample at $t=0$**; it is also consistently observed at **larger timesteps**, corresponding to noisier states earlier in the reverse process. In particular, the gain is often strongest at **intermediate-to-high noise levels**. For example, at training step 31k, the improvement increases from `-0.0096` at $t=0$ to `-0.1090` at $t=800$. This pattern would be difficult to explain if the method were merely acting as a late output correction applied near the end of sampling.
> > >
> > > This is exactly the role we intend REPA-P to play. We do not claim that every hidden feature should literally satisfy the PDE. Rather, the alignment encourages intermediate representations to remain decodable into a physically plausible state, so that the denoiser relies on more physics-consistent internal computation throughout the reverse process. If the effect came only from stronger endpoint supervision, one would expect the improvement to be concentrated only near the final clean state. Empirically, however, the benefit is visible across a broad range of timesteps and is often most pronounced away from $t=0$, especially in the middle of the trajectory.
> > >
> > > We have added this timestep-wise analysis and corresponding discussion to the revised paper to clarify why intermediate alignment helps beyond final-output matching.
> > >
> > > All reviewer-suggested revisions have been incorporated into the paper.

---

### Official Review · Reviewer_VZ8F · 2026-03-10

**Soundness:** 2
**Presentation:** 2
**Significance:** 2
**Originality:** 2
**Overall Recommendation:** 4
**Confidence:** 3

**Summary:**

This work build upon the Physics-Informed Diffusion Model (PIDM Bastek et al., 2025). Rather than enforcing physical constraints only at the final output, they supervise the intermediate states with PDE residuals. This provides in-situ gradients that shorten the credit assignment path and compel the network to “think” in physics, ensuring that latent features (from the bottleneck of the U-Net backbone) align with physical principles before the final decoding stage. In this way they do not impose the PDE constrains  only at the output but also force through the objective function to internalise the constrains within the latent features. They experiment on three PDE-governed tasks: Darcy flow (steady state), Topology optimization (structural compliance), and Electrostatic charge potential (Poisson equation).

**Compliance With Llm Reviewing Policy:**

Affirmed.

**Final Justification:**

Their answers address my questions, and I hope they will incorporate them into a revised version of the paper. Therefore, I am raising my score.

**Key Questions For Authors:**

1. How does the proposed method performs compare to [PG-Diffusion, CoCoGen, PIDM] and  a standard Diffusion model?
2. How do you interpret the negative percentage in compliance error CE? (Figure 3)


[PG-Diffusion]: Shu et, al., A physics-informed diffusion model for high fidelity flow field reconstruction

[CoCoGen]: Jacobsen et al., CoCoGen: Physically-Consistent and Conditioned Score-based Generative Models for Forward and Inverse Problems

[PIDM ]:  Bastek et al., Physics informed diffusion models

**Limitations:**

yes

**Strengths And Weaknesses:**

- Originality  & Significance : The paper builds upon previous work that trains denoising diffusion models with embedded physical constraints (Bastek et al, 2025). The main idea of the paper is to apply supervision not only to the final output (Bastek et al, 2025) but also to  internalize  the physical constraints  within the diffusion backbone, which is interesting.
- Soundness: The paper is technically sound but the experiments do not support it. The results are compared only to one baseline. Moreover it is not clear what do  they use as baseline. I assume they use a standard Diffusion model or Physics-Informed Diffusion Model (Bastek et al., 2025). In the caption of Figure 2 is described as baseline diffusion.
- Presentation: There are errors in the experiment section. In the text (p.5) is mentioned that is reported RMAE, for the Darcy flow additionally MSE,  for topology optimization additionally CE and VFE and for the charge potential problem the physics loss. Table1 for Topology optimization reports CE and  the physics loss, for Darcy flow  data (?) and physics loss and for the change data (?) and physics loss.  Table 2 reports physics loss, CE and VFE for the Topology optimization  experiment.

---

> ### Author Rebuttal · Authors · 2026-03-30
>
> We sincerely thank the reviewer for the careful reading and for highlighting two main issues: **(1) unclear or insufficient baselines** and **(2) inconsistencies in the experimental presentation**. We are grateful for these precise comments and address them below.
>
> ### 1. What is the baseline in the paper?
>
> We thank the reviewer for pointing out that this was unclear in the submission. We agree that this was our fault.
>
> The **main baseline in the original paper** is **PIDM / output-only physics-informed diffusion**, namely the same diffusion backbone trained with **physics residual supervision only at the final output**, without the intermediate projection heads used in our method. Our goal was to isolate the effect of **where** physics is imposed: **output only (PIDM)** versus **output + intermediate representations (ours)**.
>
> However, in the caption of Figure 2, we used the phrase **“baseline diffusion”**, which was ambiguous and could refer to either:
> - a **standard diffusion model without physics**, or
> - the **PIDM-style output-only physics-informed model**.
>
> We thank the reviewer for catching this ambiguity. In the revised manuscript, we now clearly distinguish:
> - **Standard diffusion**: no explicit physics constraint
> - **PIDM / output-only baseline**: physics imposed only on the final output
> - **REPA-P (ours)**: output-level physics plus intermediate representation alignment
>
> We hope this revision makes the baseline definition much clearer.
>
> ### 2. Comparison to standard diffusion, DiffusionPDE, PG-Diffusion, CoCoGen, and PIDM
>
> We thank the reviewer for emphasizing the need for broader comparisons. We fully agree that comparing to only one baseline is insufficient, and we have expanded the experiments accordingly.
>
> For the **Darcy** task, we now include:
> - **Standard diffusion**
> - **PIDM / output-only physics-informed diffusion**
> - **PG-Diffusion**
> - **DiffusionPDE**
> - **CoCoGen**
> - and their corresponding **+ REPA-P** variants
>
> To ensure fairness, all baselines are trained and evaluated on the **same Darcy dataset**, under the **same task definition**, with the **same 64×64 setting**, and using a **unified evaluation protocol**.
>
> Under this unified setting, the original baselines achieve the following results:
>
> | Method | PSNR | Physics residual |
> |---|---:|---:|
> | **PG-Diffusion** | `35.8968` | `0.1041` |
> | **DiffusionPDE** | `36.4928` | `0.1136` |
> | **CoCoGen** | `33.1300` | `3.0133` |
>
> We also evaluated the corresponding **+ REPA-P** variants under the same protocol. The results show a consistent trend that adding **REPA-P** improves both reconstruction quality and physical consistency across these backbones:
>
> | Method | PSNR | Physics residual |
> |---|---:|---:|
> | **PG-Diffusion + REPA-P** | `37.2458` | `0.0734` |
> | **DiffusionPDE + REPA-P** | `37.9785` | `0.0678` |
> | **CoCoGen + REPA-P** | `36.1547` | `1.2651` |
>
> We thank the reviewer for this suggestion, which helped strengthen the empirical support of the paper. These results will be included in the revised experimental section and comparison tables, providing stronger evidence that the benefit of REPA-P is not limited to our original backbone but generalizes across multiple diffusion-based physical generative models.
> ### 3. Inconsistency between the text and Table 1 / Table 2
>
> We thank the reviewer for pointing this out.
>
> The main issue is that **Table 1** was intended as a **compact summary table**, while some task-specific metrics were deferred to later tables, but we did not explain this clearly enough. As a result, the label **“Data”** in Darcy was vague, **VFE** for topology appeared only in Table 2, and for the **charge** task the text mentioned only physics loss while Table 1 reported both **data** and **physics** metrics.
>
> In the revised manuscript, we now explicitly state that **Table 1 is a summary table of selected key metrics**, replace ambiguous labels such as **“Data”** with their exact meaning, clarify that the **charge** task reports both **data error** and **physics residual**, and cross-reference **VFE** to the detailed topology table. We thank the reviewer again for identifying this presentation issue.
>
> ### 4. Negative CE in Figure 3
>
> We thank the reviewer for catching this.
>
> The negative value in Figure 3 was caused by a **bug in the plotting/annotation script**. This issue only affected the figure display and did **not** affect the numerical evaluation tables. We have fixed the bug and corrected Figure 3.
>
> We sincerely thank the reviewer again for the precise and constructive feedback. These comments significantly improved the clarity of the experimental section.

---

> > ### Author Rebuttal · Reviewer_VZ8F · 2026-04-02
> >
> > I thank the authors for responding to my comments. Their answers address my questions, and I hope they will incorporate them into a revised version of the paper. Therefore, I am raising my score.

---

### Official Review · Reviewer_aceH · 2026-03-11

**Soundness:** 3
**Presentation:** 4
**Significance:** 3
**Originality:** 3
**Overall Recommendation:** 4
**Confidence:** 3

**Summary:**

Usually, physics-informed diffusion models only enforce physics in the output. This work addresses this limitation by introducing a teacher-free physics-informed representation alignment in the latent space. It decodes hidden activations into physical states and applies PDEs and conditions on the intermediate representations. This is done with heads that are discarded when inferencing. Experiments demonstrate the improved fidelity of the trained models.

**Compliance With Llm Reviewing Policy:**

Affirmed.

**Final Justification:**

The rebuttal addressed my main concerns, and I appreciate the clarifications provided. The rebuttal increased my confidence in the paper’s soundness and helped resolve the key issues. However, the overall balance of strengths and weaknesses is still consistent with my original assessment, so my final score remains weak accept.

**Key Questions For Authors:**

How sensitive is performance to projection head placement?

**Limitations:**

Yes

**Strengths And Weaknesses:**

+Simple idea for a well-motivated problem

+Works across multiple PDE tasks

+Low parameter overhead

-Experiments limited to 2D PDEs

-Rather practical contributions, less theoretical

-Extra training heads add training complexity.

---

> ### Author Rebuttal · Authors · 2026-03-30
>
> We sincerely thank the reviewer for assessing our paper and highlighting the simplicity, motivation, low parameter overhead, and cross-task effectiveness of our method. We also appreciate the thoughtful questions on projection head placement, theory, training complexity, and 2D PDEs.
>
> ## 1. Sensitivity to Projection Head Placement
>
> Thank you for this important question. Our results show that REPA-P is **sensitive to head placement in a structured way, but not brittle**.
>
> Across our ablations, **bottleneck / middle-layer alignment** gives the most robust performance, although other positions can be competitive for specific metrics. In Darcy flow, encoder alignment is competitive in PSNR, while bottleneck alignment gives the best overall balance and strongest physics consistency after tuning. In topology optimization and electrostatic charge potential, bottleneck alignment also performs best.
>
> Overall, placement matters, and the **bottleneck is the most reliable choice**. We have clarified this in the revision.
>
> ## 2. Why Placement Matters
>
> We agree that the paper is mainly methodological, but the design is not purely heuristic.
>
> Let $h_l$ be an intermediate representation and let $x_0 = g_l(h_l)$ be the final prediction. If physics is imposed only at the output, the gradient reaching $h_l$ can be written as $dL/dh_l = (1/\sigma^2(t)) J_{g_l}(h_l)^T J_R(x_0)^T R(x_0)$.
>
> Thus, the supervision must pass through the **remaining backbone**, which can weaken or poorly condition the physics signal.
>
> With REPA-P, we decode the intermediate feature through a lightweight projection head,
>
> $$ z_l = \Pi_l(h_l). $$
>
> We then impose physics directly on $z_l$, so the gradient becomes $dL/dh_l = (1/\sigma^2(t)) J_{\Pi_l}(h_l)^T J_R(z_l)^T R(z_l)$.
>
> This provides a **shorter and better-conditioned supervision path**. It also explains the empirical trend: very early features are often too local, output-only supervision is too delayed, and bottleneck features best balance semantic abstraction and direct physical supervision.
>
> We have clarified this intuition in the revision.
>
> ## 3. On the Theoretical Contribution
>
> We agree that this is not a theory paper in the strict sense, but the method is supported by two principled views.
>
> First, from a **virtual-observation / likelihood** perspective, physics residual supervision corresponds to enforcing a virtual observation that the PDE residual should be zero, which under a Gaussian model gives $L_{phys}(t) = \frac{1}{2\sigma^2(t)} \|R(\cdot)\|_2^2$.
>
> Second, from a **gradient-flow / credit-assignment** perspective, applying physics supervision at intermediate layers shortens the supervision path and helps reduce shortcut learning compared with output-only supervision.
>
> We have revised the manuscript to make these motivations more explicit.
>
> ## 4. Extra Heads and Training Complexity
>
> We appreciate this concern. The additional heads are intentionally lightweight, used **only during training**, and **discarded at inference time**. Therefore, **inference-time parameter count is unchanged**, **inference-time sampling cost is unchanged**, and the extra cost is limited to modest training-time overhead from small projection heads and residual evaluations.
>
> We have made the distinction between training-time and inference-time cost clearer in the final version.
>
> ## 5. Limitation to 2D PDEs
>
> We fully agree that our current experiments are limited to 2D PDE benchmarks. Our goal was to first validate the mechanism in a controlled setting where the governing equations are known and the effect of representation alignment can be isolated clearly.
>
> That said, the framework itself is **not inherently restricted to 2D**. REPA-P only requires (1) an intermediate representation that can be projected into the target physical state space, and (2) a differentiable residual operator defined on that space.
>
> These assumptions are not specific to 2D, so in principle the framework can extend to **3D PDEs**, **higher-dimensional physical fields**, and other discretizations when the residual operator is available. We have clarified that the current 2D scope is an **evaluation limitation**, not a conceptual one.
>
> We are grateful for these comments and believe they have further improved the paper's clarity and positioning.

---

> > ### Author Rebuttal · Reviewer_aceH · 2026-04-02
> >
> > Thansk for the extensive rebuttal. My concerns have been resolved. However, I think the overall contributions of the paper are already reflected in my score.

---

### Official Review · Reviewer_6VyA · 2026-03-12

**Soundness:** 3
**Presentation:** 3
**Significance:** 3
**Originality:** 3
**Overall Recommendation:** 4
**Confidence:** 3

**Summary:**

The authors propose to use REPA-like regularization of intermediate representations in physics-informed diffusion models to avoid shortcut solutions which fit training statistics yet generalize poorly under shifted boundary conditions. The proposed method, REPA-P, is a teacher-free alignment framework that uses first-principles residuals as supervision. REPA-P uses projection heads during training to regularize the early-/mid-layer representations and discards them during inference. Empirical results are shown across three 2D scientific benchmarks (Darcy flow, topology optimization, and electrostatic charge potential). The authors claim faster convergence and improved out-of-distribution robustness.

**Compliance With Llm Reviewing Policy:**

Affirmed.

**Final Justification:**

I appreciate the authors' detailed and thoughtful rebuttal, which has satisfactorily addressed my questions. I have no further concerns at this point. I will keep my current score, as my initial rating was intentionally set on the higher side and already reflected my view that the post-rebuttal paper is in the borderline accept range.

**Key Questions For Authors:**

1. See weaknesses.
2. Regarding the ablation study tables (Table 2-4), would it be beneficial to combine them into a single table by joining them column-wise? It might give readers an easier time to find results as well as save some space. I notice there are some other interesting results in the Appendix that might be moved to the main text if space permits.

**Limitations:**

Yes

**Strengths And Weaknesses:**

Strengths:
1. The idea is well motivated and fairly straightforward. The introduction of “physical decodability” also reasonable. It is also well conveyed by Figure 1, although the figure style might be overly fancy/flashy.
2. The quantitative gain compared to the baseline diffusion U-Net is quite decent and obvious. The ablation study is fairly comprehensive. The per-layer physics residual analysis (Figure 5) is interesting.

Weaknesses:
1. As a general suggestion on the Experiments section, a lot of implementation details can be left in Appendix, leaving space for more explanations on some brief background introduction and explanation of the specific experiments. For example, as a person without prior knowledge of Darcy flow, it is not very easy for me to understand what the model is trying to do, how well it does the objective, etc., when looking at Figure 2.
2. Despite the comprehensive ablation study, the comparison to a single baseline is a little weak. I would suggest adding something simple, such as diffusion transformer (DiT) and DiT + REPA-P. In addition, are there other physics-informed regularizations that can be compared against?

---

> ### Author Rebuttal · Authors · 2026-03-30
>
> We sincerely thank the reviewer for their careful reading and constructive feedback. We especially appreciate the positive comments on our method's motivation, the concept of **physical decodability**, the clarity of the core idea, and the value of the ablation and per-layer analyses. We respond to each point below.
>
> ## 1. Experimental Presentation and Benchmark Background
>
> We agree that the experimental section can be made more accessible for readers less familiar with Darcy flow or topology optimization. We therefore revised it as follows:
>
> - We moved low-level implementation details to the appendix, allowing the main text to focus on task definitions, model predictions, and metric interpretation.
> - We added a brief task-oriented introduction for each benchmark. For Darcy flow, for example, we now explicitly state the governing PDE and clarify the meaning of the residual metric.
> - We revised the figure captions to better explain the qualitative plots and the connection between lower residual maps and stronger physical consistency.
>
> We believe these changes improve the readability of the experimental section.
>
> ## 2. Visual Style of Figure 1
>
> Thank you for this helpful suggestion. We agree that the original Figure 1 can be simplified to better match the academic style of the paper, and we redesigned it accordingly:
>
> - We replaced the high-contrast multi-color styling with a more unified and muted palette.
> - We simplified the connectors and improved the alignment among the backbone, projection heads, and physics operator blocks.
> - We reorganized the training-objective panel to make the role of each loss term clearer.
>
> The updated figure preserves the framework's core intuition while presenting it in a cleaner and more technical style. Due to rebuttal-phase policy, we cannot upload the revised manuscript at this stage, but the updated figure has been prepared for the final revision.
>
> ## 3. Comparison to Stronger Baselines, Especially DiT
>
> We appreciate this suggestion and agree that comparison with a stronger backbone is important. Following the reviewer's recommendation, we added **DiT-based experiments** to test whether REPA-P generalizes beyond U-Net to transformer-based diffusion backbones.
>
> ### (a) DiT on Darcy Flow
>
> On Darcy flow, **middle-layer alignment** reduces the physics residual from the DiT baseline of `0.0719` to `0.0534`, while **output-level supervision** achieves an even stronger result of `0.0304`.
>
> These results show that:
>
> - REPA-P is **not limited to U-Net** and remains effective on transformer-based diffusion backbones.
> - The **optimal alignment position depends on the backbone**, further supporting our central message that **where physics is injected matters**.
>
> ### (b) DiT on Mechanics / Topology Optimization
>
> On the mechanics/topology optimization task, all error metrics improve substantially. **Middle-layer alignment** performs best, reducing **CE** from `14.93%` to `6.79%` and **VF** from `16.43%` to `7.59%`, while keeping the physics residual at the same order of magnitude as the baseline.
>
> Overall, these experiments show that REPA-P is **not tied to a single diffusion architecture**, and that **intermediate representation alignment** is a generalizable and meaningful design dimension rather than a U-Net-specific trick. We thank the reviewer for encouraging us to include this stronger baseline.
>
> ## 4. Comparison to Other Physics-Informed Regularizations
>
> We agree that this is an important question. Our main controlled comparison is with **output-only physics supervision**, because our central question is not simply whether physics regularization helps, but whether applying physics constraints to **intermediate representations** provides additional benefits beyond enforcing them only at the output.
>
> For this reason, output-only supervision is the most direct baseline for isolating the effect of **alignment position**. We will make this motivation more explicit in the revised manuscript and expand the discussion of other physics-informed regularization approaches, supported in part by the newly added **DiT + output-only** results.
>
> ## 5. Organization of the Ablation Tables
>
> Thank you for this practical suggestion. We streamlined the ablation presentation so that a compact summary remains in the main text, while the full detailed ablations are moved to the appendix.
>
> This improves readability and frees up space in the main text for benchmark background and experiment explanations, directly addressing the reviewer's first concern.
>
> Thank you again for your valuable review.

---

> > ### Author Rebuttal · Reviewer_6VyA · 2026-04-01
> >
> > I appreciate the authors' detailed and thoughtful rebuttal, which has satisfactorily addressed my questions. I have no further concerns at this point. I will keep my current score, as my initial rating was intentionally set on the higher side and already reflected my view that the post-rebuttal paper is in the borderline accept range.

---

### Decision · Program_Chairs · 2026-04-30

**Decision:**

Accept (regular)

**Comment:**

The reviewers agreed that the core idea of enforcing "physical decodability" by applying PDE constraints to the intermediate layers of a diffusion model is intuitive, well-motivated, and elegantly simple.
The proposed REPA-P framework effectively tackles the "shortcut learning" problem often observed in physics-informed diffusion models, and it does so without introducing any computational penalty during inference, as the projection heads are discarded post-training. Reviewers mentioned the clear and consistent empirical gains, noting that the method successfully reduces physics residuals, accelerates convergence, and improves out-of-distribution robustness to boundary-condition shifts across the evaluated scientific benchmarks.

During the initial review phase, reviewers raised concerns regarding the scope of the evaluation, specifically noting the reliance on a single baseline, the exclusive use of the U-Net architecture, and the relative simplicity of the 2D PDE datasets.
Reviewers also asked for a stronger theoretical justification as to why intermediate latent constraints benefit the denoising process.
The authors provided a rebuttal that addressed these weaknesses. They significantly expanded their baselines (adding PG-Diffusion, CoCoGen, and DiffusionPDE), implemented a Diffusion Transformer (DiT) architecture to demonstrate architectural generalizability, and introduced a harder benchmark of turbulent channel flow.
Furthermore, they addressed theoretical concerns by providing gradient-flow intuition and a detailed timestep-wise analysis demonstrating that intermediate alignment improves the internal denoising trajectory.
While one reviewer maintained a borderline reject score due to a philosophical reservation that strict physical constraints might act as a suboptimal inductive bias and limit generative diversity at scale, the broader consensus found that the authors addressed the empirical and methodological criticisms.

The comprehensive discussion and rebuttal process significantly strengthened the manuscript. The reviewers note that the additions promised during this phase are critical to the completeness and clarity of the work. Specifically these include: 1) the clarified definitions distinguishing the standard diffusion, PIDM (output-only), and REPA-P baselines; 2) the expanded baseline comparisons (PG-Diffusion, CoCoGen, DiffusionPDE); 3) the architectural ablation results using the DiT models; 4) the inclusion of the turbulent channel flow benchmark; 5) the explicit gradient-flow intuition and the timestep-wise denoising trajectory analysis; and 6) corrections to the labels and plots as discussed. These additions collectively provide a robust foundation for evaluating the proposed method.